# Biology Instructions: A Dataset and Benchmark for Multi-Omics Sequence Understanding Capability of Large Language Models

## Abstract

Large language models have already demonstrated their formidable capabilities in general domains, ushering in a revolutionary transformation. However, exploring and exploiting the extensive knowledge of these models to comprehend multi-omics biology remains underexplored. To fill this research gap, we first introduce Biology-Instructions, the first large-scale multi-omics biological sequences-related instruction-tuning dataset including DNA, RNA, proteins, and multi-molecules, designed to bridge the gap between large language models (LLMs) and complex biological sequences-related tasks. This dataset can enhance the versatility of LLMs by integrating diverse biological sequenced-based prediction tasks with advanced reasoning capabilities, while maintaining conversational fluency. Additionally, we reveal significant performance limitations in even state-of-the-art LLMs on biological sequence-related multi-omics tasks without specialized pre-training and instruction-tuning. We further develop a strong baseline called ChatMultiOmics with a novel three-stage training pipeline, demonstrating the powerful ability to understand biology by using Biology-Instructions. Biology-Instructions and ChatMultiOmics are publicly available and crucial resources for enabling more effective integration of LLMs with multi-omics sequence analysis.

## 1 Introduction

Understanding the complex activities across various omics in living organisms is of paramount importance. This includes studying DNA regulatory elements that control gene expression (Emilsson et al., 2008), RNA regulation (Mattick, 2004) that influences protein synthesis, and the functional properties of proteins themselves (Marcotte et al., 1999). These molecular processes critically affect the development of diseases and the synthesis of drugs within organisms. Recent BERT-like encoder-only models (Devlin, 2018) have achieved significant advances in natural language understanding tasks. When applied to genome or protein understanding tasks, these models (Zhou et al., 2023; Rives et al., 2021) are capable of capturing complex intrinsic relationships within biological sequences, achieving high accuracy in tasks such as promoter prediction. However, their reliance on specific classification or regression heads to predict a single task at a time limits their versatility, and their repeated fine-tuning sessions with different prediction heads to address multiple tasks further complicate the training, inference, and deployment process.

In contrast, powerful general-purpose large language models (LLMs) such as GPT-4 (Achiam et al., 2023) and Gemini (Achiam et al., 2023; Team et al., 2023) based on vast amounts of natural language tasks and data that encompass the general knowledge system of humanity, have shown substantial potential in domain-specific tasks. These decoder-only models approach every task as a completion task through next-token prediction, and offer an alternative by integrating various biological sequence-related tasks using natural language as an intermediary while retaining conversational capabilities. Therefore, utilizing LLMs combined with unified training and dataset construction techniques can make it possible to replace BERT-like models with the complicated fine-tuning pipeline.

Recently, some studies have explored leveraging LLMs for tasks related to biological sequences through instruction tuning, such as ChatNT (Richard et al., 2024) and ProLlama (Lv et al., 2024). Although showing promising results, these models are trained on instruction-tuning datasets con-

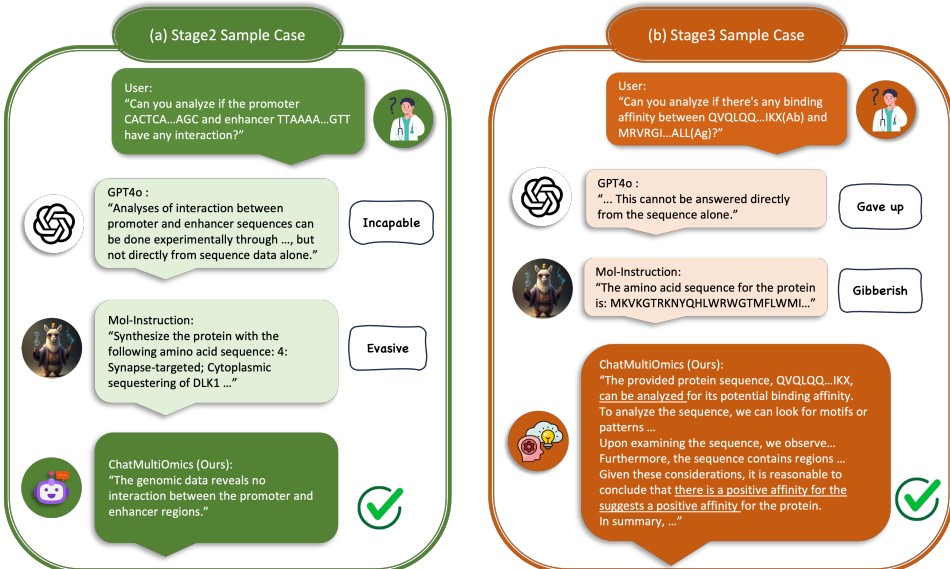

Figure 1: Comparative examples showcasing ChatMultiOmics performance against baseline models on multi-molecular tasks. (a) shows an example from Enhancer-Promoter Interaction Prediction task (Min et al., 2021) after stage2 training. (b) shows an example from Antibody-Antigen Neutralization (AAN) task (Zhang et al., 2022) after stage3 training. Note that AAN data is not included in stage3 training, which showcases our model's task generalization capability.

taining only basic language patterns, underutilizing the full linguistic capabilities of the original LLMs. Moreover, these models mainly focus on single-omics data for either protein or DNA, limiting their potential to provide important multi-omics understanding ability as a unified foundational language model. Inspired by multimodal LLMs like MiniGPT-4 (Zhu et al., 2023), we see an opportunity to extend this approach to biology. In biology, where molecular interactions are fundamentally grounded in the central dogma (Crick, 1970), integrating multi-omics data holds immense potential for generating mutually reinforcing insights.

Our study attempts to answer a key question: can instruction-tuned language models, proficient in understanding human language, also excel in understanding biological sequences to address biologically critical tasks? The motivation behind this inquiry lies in the intrinsic parallels between biological sequence data and human language—both are discrete, sequential, abundant and rich in encoded information. These shared characteristics suggest that, with appropriate adaptation, instruction-tuned LLMs could unlock transformative capabilities in biology.

To properly investigate the gap between human language and biological sequences understanding, we introduce Biology-Instructions, the first large-scale, multi-omics biology sequence-related instruction-tuning benchmark supporting 21 distinct tasks. This benchmark covers DNA, RNA, proteins and multi-molecular prediction tasks for a comprehensive understanding of biology. With Biology-Instructions, we conduct a comprehensive evaluation of kinds of open-source and closed-source LLMs, and reveal that most models including the state-of-the-art GPT-4o, perform at near-random levels on biological sequence-related understanding tasks without prior specialized training. This suggests the lack of inherent biological sequence knowledge in LLMs and highlights the need for methods to effectively integrate these tasks with LLMs.

Furthermore, we attempt to activate the biological multi-omics sequence understanding ability of LLMs with the constructed instruction data. We discover that solely performing instruction tuning on Biology-Instructions can not yield satisfactory results. To address this gap, we propose a three-stage training pipeline: (1) train the model on unsupervised DNA, RNA, and protein sequences; (2) train the model on the question-answer pairs of Biology-Instructions; (3) train the model on reasoning data. The first stage serves as a warm-up to enhance the model's ability to understand biological sequences. In the second stage, the model follows natural language instructions to interpret biological sequences. In the third stage, the model leverages the implicitly learned knowledge base to perform reasoning and deepen its understanding of biological sequences. We include rea-

soning data that starts with biological sequence analysis and concludes with results based on prior analyses and reasoning. This approach ensures that models maintain comprehensive conversational abilities while gaining deeper insights into biological sequences and tasks. We have implemented this training pipeline on Llama3.1-8b-Instruct (Dubey et al., 2024) using Biology-Instructions, resulting in significant performance improvements shown in Figure 1. Our findings and experiences are thoroughly documented. Our contributions can be summarized as:

- **Multi-omics Instruction-Following Data.** We present the first dataset specifically designed for multi-omics instruction-following, which includes reasoning instruction data and multi-sequence, multi-molecule instruction data. This dataset aims to improve the ability of LLMs to comprehend and analyze biological sequences.

- **Multi-omics Instruction-Following Benchmark.** We benchmark Biology-Instructions on open-source and closed-source LLMs. Our results reveal that even current LLMs can not solve biological sequences-related tasks.

- **Biology-Specific LLMs and Three-Stage Training Pipeline.** We develop a biology-focused LLM capable of handling tasks related to multi-omics sequences by training an open-source LLM on biology-specific instructions. We propose an efficient and novel three-stage pipeline to enhance the biology learning ability of LLM based on some important findings.

- **Fully Open-Source.** We will release three assets to the public: the Biology Instructions dataset, the entire training pipeline's codebase and the model checkpoints. The Biology-Instructions is publicly available through an anonymous data link.[1] .

## 2 RELATED WORKS

### 2.1 LARGE LANGUAGE MODELS

In recent years, LLMs have demonstrated significant advancements in the field of natural language processing (NLP). These models undergo self-supervised training on a substantial corpus of data in order to acquire knowledge. By means of fine-tuning the instructions, the capabilities of the model are enhanced, enabling it to respond to questions based on the specific prompt. Currently, numerous open-source models are available, including the Llama series (Dubey et al., 2024), Qwen series (Bai et al., 2023), GLM series (GLM et al., 2024), and numerous models fine-tuned based on Llama, such as Alpaca (Taori et al., 2023) and Vicuna (Chiang et al., 2023). Additionally, Galactica (Taylor et al., 2022) is a model that demonstrates exceptional performance in scientific domains and is trained on data from a multitude of scientific fields. Furthermore, there are closed-source SOTA models, such as GPT-4o and GPT-4o-mini. However, these models are not pre-trained on specific biological data, and their capabilities are severely constrained, even Galactica.

### 2.2 BIOLOGY LARGE LANGUAGE MODELS

Researchers have concentrated on enhancing the capabilities of LLMs in the biology area. Instruct-Protein (Wang et al., 2023) aligns human and protein language through knowledge instructions. Another study (Fang et al., 2023) utilizes the protein part of a specially designed dataset called Mol-Instructions for instruction tuning with LLaMA-7B. ProLLaMA (Lv et al., 2024) is also a recent work focusing on multi protein tasks through a two-stage traing process from LLaMA-2. These methods can only deal with several protein tasks well, limited by fixed instruction templates. BioMedGPT (Zhang et al., 2023) is equiped with special vision encoder, allowing the model to answer multi-modal biological questions. However, lack of specialized large-scale biological instruction datasets, BioMedGPT cannot understand biological sequence languages very well. ChatNT (Richard et al., 2024) integrates a biological sequence encoder with a LLM, enabling effective handling of DNA-centric tasks using only an instruction-tuning dataset. However, it faces challenges in combining multiple encoder models from various omics domains into a unified LLM due to dependence on the encoder's capabilities.

---

[1]https://anonymous.4open.science/r/Biology-Instructions-FD66/

Table 1: Comparing with baseline large language models (LLMs). We employ general and domain-specific LLMs both. We report the number of parameters, expertises and access of them. Params means the number of parameters.

| Model | Params | Expertise | Access |
|---|---|---|---|
| *General LLMs* | | | |
| Llama3.1-Instruct | 8B | General tasks | Open |
| GLM4 | 9B | General tasks | Open |
| Qwen2 | 7B | General tasks | Open |
| Alpaca | 7B | General tasks | Open |
| Vicuna-v1.5 | 7B | General tasks | Open |
| Galactica | 1.3B | General scientific tasks | Open |
| Llama2-chat | 7B | General tasks | Open |
| GPT-4o | - | General tasks | Close |
| GPT-4o-mini | - | General tasks | Close |
| *Domain-specific LLMs* | | | |
| InstructProtein | 1.3B | Protein tasks | Open |
| LLama-molinstruct-protein | 7B | Protein tasks(mainly) | Open |
| BioMedGPT | 7B | Protein and DNA tasks | Open |
| ProLLaMA | 7B | Protein tasks | 1-stage Open |
| ChatNT | 7B+500M | DNA tasks(mainly) | Not released yet |
| Our ChatMultiOmics | 8B | Multi-omics tasks | Open |

## 3 BIOLOGY-INSTRUCTIONS

### 3.1 OVERVIEW OF BIOLOGY-INSTRUCTIONS

To build a large-scale biology instruction-following dataset, we have gathered biology sequence data from a substantial aggregation of sources. This effort has resulted in a dataset encompassing 21 subtasks related to multi-omics fields. The Biology-Instructions exhibits the following characteristics:

**Multi-omics** Biology-Instructions comprises 21 subtasks across three types of omics, including single-omics tasks and multi-omics interaction tasks. Joint training of different omics not only enhances efficiency by accomplishing multiple omics tasks with a single model but also improves the model's capability in a specific omics domain.

**Large Scale** With over 3 million training samples, the Biology-Instructions dataset provides an extensive foundation for biological sequences-related instruction data. This large-scale dataset enables models to better understand the traits and functions of biological sequences, leading to more accurate and comprehensive responses to given questions.

**High Quality** To ensure the quality of the dataset, we manually draft question and answer templates for each task type and expand the template pool using Cluade-3.5-sunnet and GPT-4o. The resulting number of question-answer template pairs for each task range from 10,000 to 100,000, depending on the data magnitude of each task type. Throughout this process, we emphasize the importance of diversity in grammar and language style, ensuring that samples in the Biology-Instructions dataset have different question-answer style. For examples of question-answer template pairs, please refer to Table 10.

**Reasoning data** Although previous studies (Richard et al., 2024; Liu et al., 2024b; Lv et al., 2024) have demonstrated large-scale primary instruction-following datasets can teach LLMs to answer biological sequences-related questions, they often fail to fully harness the powerful language abilities of LLMs, as they focus primarily on basic language patterns. In other words, they failed to leverage the powerful conversational abilities of these models to form natural and fluent dialogues, and further utilize reasoning to enhance the validity of the output results. To address this limitation, we design a prompt that requires powerful closed-source LLMs to reformulate answers for a subset of Biology-Instructions' validation set and provide polished answers ready for end-users to read and understand, based on given questions and original answers. We encourage the model to deeply analyze the sequence and question first and then generate a final polished answer grounded in previous analysis and reasoning.

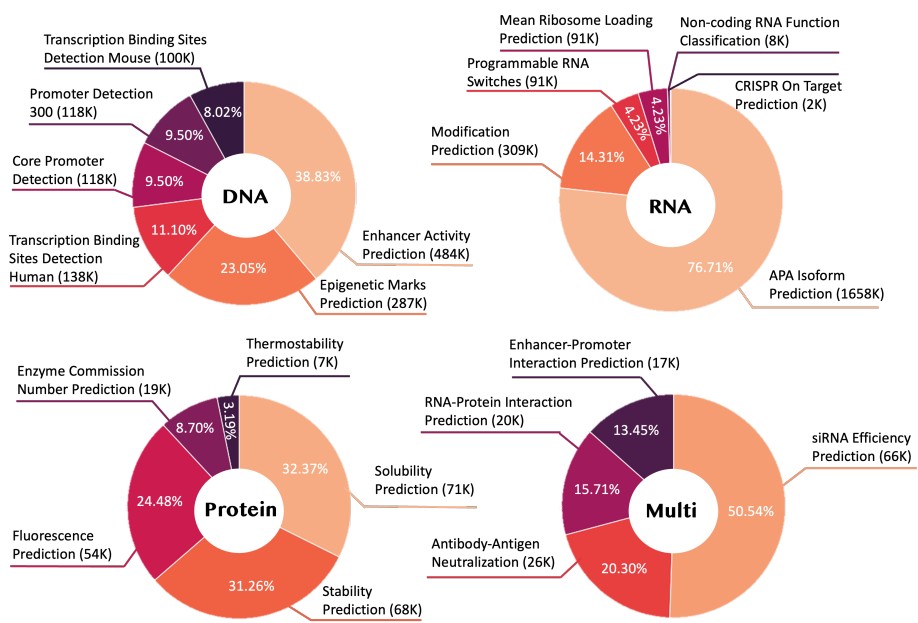

Figure 2: Distribution of tasks across four omics types in our dataset.

## 3.2 BIOLOGY-INSTRUCTIONS CONSTRUCTION

### 3.2.1 TASKS

As presented in Figure 2, the Biology-Instructions dataset comprises 21 tasks: 6 DNA tasks, 6 RNA tasks, 5 protein tasks, and 4 multi-molecule tasks. When considering the number of input sequences, there are 4 multi-molecule interaction tasks and 17 single-molecule tasks. Tasks were sourced from high-impact literature, journals, and competitions, ensuring coverage of biologically critical aspects in structure, function, and engineering across DNA, RNA, proteins, and their interactions. We focus on predictive sequence-understanding tasks, leaving generative applications, such as sequence design, for future research. To the best of our knowledge, Biology-Instructions is the first instruction dataset to include multi-omics tasks and multi-molecule interaction tasks. For detailed task definitions and distribution, please refer to Appendix A.2.

### 3.2.2 TEMPLATES

To convert the original classification and regression task dataset into an instruction tuning dataset, we employ question-answer templates to integrate the data. The primary objective of creating these templates is to teach the model how to follow biological instructions and complete tasks without overfitting to specific language patterns. To achieve this, we prioritize diversity in language styles, tones and lengths during the template construction process. We manually constructed 10 question templates and 10 answer templates for each task, covering various styles including, but not limited to, request, concise, informal, and academic styles. Then, we used GPT-4o and Claude-3.5-sunnet to expand the templates. Depending on the data volume for each task, we included 100 to 300 question templates and 100 to 300 answer templates. Ultimately, each task resulted in 10,000 to 100,000 question-answer template pairs. Since biological sequences are generally much lengthier than natural language prompts, we place the biological sequence at the very beginning of question templates for single biology sequence tasks for non-interaction tasks. This approach helps prevent the prompts from being overwhelmed by the lengthy biological sequences, ensuring that the model can accurately understand the question and complete the task. Figure 3 provides examples of the instruction prompts constructed for each type of omics, illustrating the diversity and structure of the templates used in the dataset.

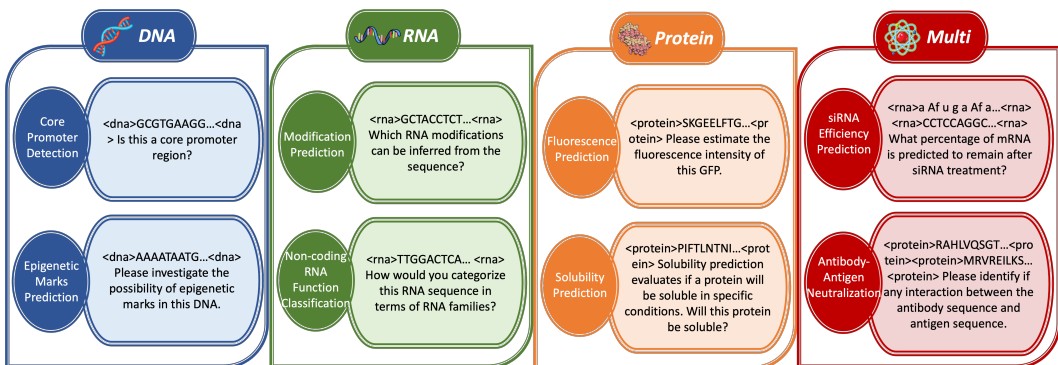

Figure 3: Examples of instruction prompts constructed for each omics type.

### 3.2.3 REASONING DATA CONSTRUCTION

Similar to the data construction method used by LlaVA (Liu et al., 2024a). For a biology sequence $X_s$ and its related question $X_q$, simple answer $Y_s$, we prompt GPT-4o-Mini to construct optimized answer $Y_o$ base on the given information. Generally, the instruction data were transformed to the format USER:$X_s$, $X_q$ ASSISTANT:$Y_o$.

In the system prompt used for GPT-4o-Mini, as shown in Figure 10, we emphasized the following key points to ensure the production of high-quality data: (1) first understand the provided biological sequence and the question; (2) analyze the biological sequence at the nucleotide or amino acid level, aiming to extract question-related information from the sequence; (3) refine the answer based on the previous analysis, including a rational explanation and a chain of thought approach, especially for complex questions; (4) list any relevant knowledge and information from reliable sources, and cite these sources appropriately; (5) return the polished answer in an end-to-end style, excluding any information from the standard answer and task hint. By following this approach, we gathered 8000 final AI-polished training data points without two multi-molecule tasks: antibody-antigen neutralization and RNA-protein interaction prediction to study transfer learning for reasoning capability. Figure 4 provides an overview of the complete construction process for Biology-Instructions, including the data collection, template construction, and reasoning data construction stages.

### 3.3 EVALUATION PIPELINE AND METRICS

Our evaluation framework is designed to assess the performance of each model's output across the diverse set of tasks included in Biology-Instructions in a robust approach. The task types, regardless of their respective omics, can be organized into single-label regression, multi-label regression, binary classification, multi-class classification, and multi-label classification, each requiring specialized evaluation metrics to capture model performance nuances. The evaluation pipeline involves pre-processing data from models' output, grouping entries by task, and then computing task-specific metrics. The metrics outcomes for reporting are all scaled by 100 and rounded to 2 decimals for enhanced readability. For detailed information on specific metrics, please refer to Appendix A.3.

## 4 MODEL

As shown in Figure 5, we train a model based on Llama3.1-8B-Instruct (Dubey et al., 2024) named ChatMultiOmics using multi-omics pre-training data and Biology-Instructions. In general, we perform a three stages training paradigm to enhance the interactive biological sequence-related chat performance of the final biology assistant. For specific training details, please refer to Appendix B.

### 4.1 STAGE 1: BIOLOGICAL SEQUENCES CONTINUED PRE-TRAINING

Although the memory savings facilitated by LoRA (Devalal & Karthikeyan, 2018) are not that obvious when optimizer states are distributed across GPUs compared with training on single GPU, LoRA can still significantly reduce training time by minimizing communication between data paral-

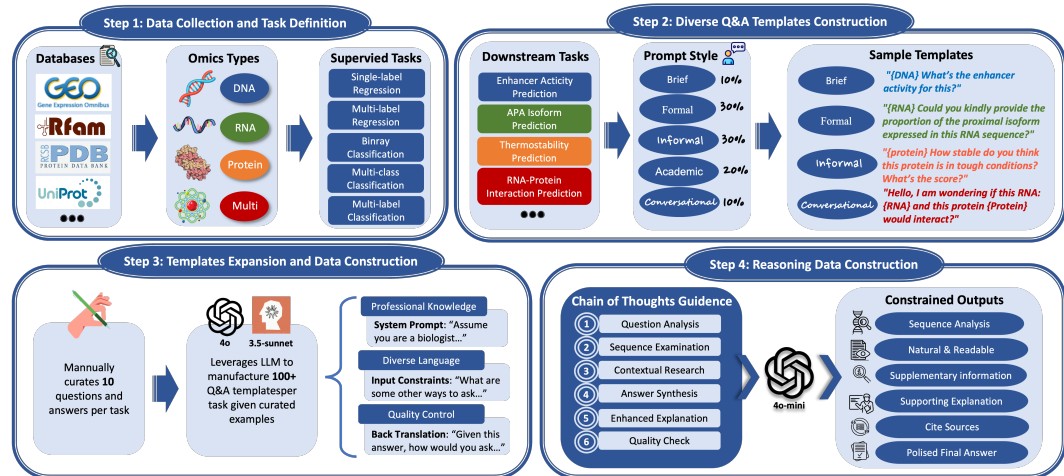

Figure 4: Overview of our data construction pipeline. Step1 shows primary databases for data collection, downstream tasks categorized by omics types, supervised tasks types involved in our benchmark. Step2 shows how we construct our instruction prompts based on downstream tasks and by diversifying prompt styles. Step3 illustrate how we leverage LLMs to augment high quality datasets. Step4 shows the key values we adhere to for reasoning data construction.

lel ranks. However, directly applying LoRA to train a chat model on Biology-Instructions results in suboptimal performance on specific downstream tasks. Specifically, the model shows near-random performance in classification and regression tasks. As noted by (Ghosh et al.), LoRA supervised fine-tuning (SFT) primarily leverages pre-trained knowledge to generate well-formed answers based on the output format learned from SFT data. We suspect that large-scale LoRA instruction tuning on biological sequence-related data suffers due to the lack of pre-training on biological sequence data, which is evident from the baseline results. Therefore, continued pre-training of the model is essential for better performance. This involves teaching the model with biological sequences to enable it to understand the nature and functions of biological sequences. For this process, we utilized unlabeled human DNA data from the Genome Reference Consortium Human genome (GRCh) (Harrow et al., 2012), human non-coding RNA data from RNACentral (rna, 2019), and protein sequences from UniRef50 (Suzek et al., 2007) during the first phase of pre-training. This initial pre-training served as a foundational warm-up to improve the model's comprehension across multi-omics biological sequences.

We employed LoRA+ (Hayou et al., 2024) for all linear layers of our model, training on a continued pre-training dataset. LoRA+ demonstrates superior convergence compared to vanilla LoRA by increasing the learning rate of the zero-initialized weight B relative to the base learning rate for normal-initialized weight A and other trainable parameters. (Hayou et al., 2024) observed that setting the learning rate of weight B to 16 times that of weight A results in more effective model convergence. However, our experiments revealed that while LoRA+ indeed improves convergence rates, applying a large learning rate multiplier can lead to instability during the continued pre-training process for biological sequences. Based on this observation, we opted for a more conservative learning rate multiplier of 4. We trained the normalization layers of the model alongside LoRA parameters.

### 4.2 STAGE 2: MASSIVE INSTRUCTION TUNING

In Stage 2, we employ the Biology-Instructions dataset, excluding the reasoning sub-dataset. In the initial attempts of training, we find that the imbalance among tasks within the dataset can pose challenges for the model in distinguishing between different tasks. To mitigate this, we randomly select 30 percent of the training data and prepend a task label in the format "[Classification/Regression:task_name]" at the beginning of each question. This method effectively aids the model in identifying different tasks and output objectives.

We use a system prompt $P_{sc}$: "You are a knowledgeable and helpful biology assistant. Please answer my biology sequence-related questions in a clear and concise manner. For regression tasks, please

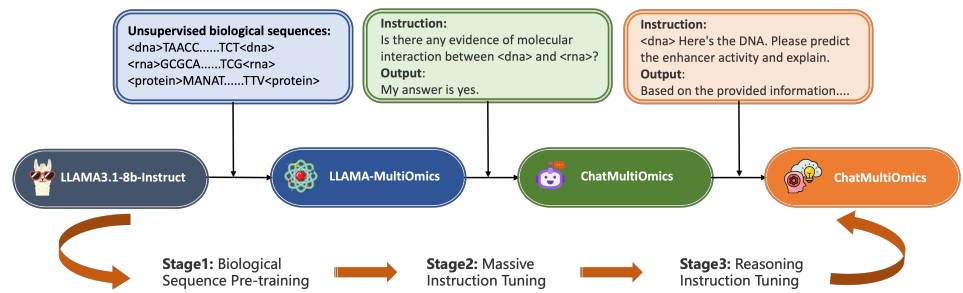

Figure 5: Overview of our three-stage training pipeline.

return a number." This prompt helps the model to differentiate biology sequence-related tasks from other tasks. As illustrated in Figure 8, we maintain the data format: SYSTEM:$P_{sc}$ USER:$X_s, X_q$ ASSISTANT:$Y_o$ consistent with the Llama3.1 instruct-tuned model chat completion format, which is crucial for optimal model performance.

### 4.3 STAGE 3: REASONING INSTRUCTION TUNING

In stage 3, we use reasoning sub-dataset from Biology-Instructions to fine-tune the model. To keep the classification and regression performance of the model, we additionally select 3000 samples from validation set composed of non-reasoning data to be trained simultaneously.

To better control the behavior of the model, a more detail system prompt $P_{sd}$ was used for reasoning data: "You are a highly knowledgeable AI assistant specializing in biology, particularly in sequence-related topics. Your primary task is to provide clear, accurate, and comprehensive answers to biology questions. When analyzing and interpreting sequences, ensure to provide step-by-step explanations to make your responses natural and easy to understand. Engage with the user by asking clarifying questions if needed and offer detailed insights into the biological sequences." In this case, the format of training sample of reasoning data is transformed to SYSTEM:$P_{sd}$ USER:$X_s, X_q$ ASSISTANT:$Y_o$.

## 5 RESULTS

### 5.1 EXPERIMENTAL SETUPS

To evaluate the biological sequence understanding capabilities of current LLMs and determine if our method can enhance LLMs performance, we compare ChatMultiOmics with various open-source general-purpose LLMs: Llama3.1-8B-Instruct (Dubey et al., 2024), Llama2-7B-Chat (Touvron et al., 2023), Alpaca-7B (Taori et al., 2023), Vicuna-v1.5-7B (Chiang et al., 2023), Qwen2-7B (Bai et al., 2023), GLM4-9B-Chat (GLM et al., 2024), and Galactica-1.3b (Taylor et al., 2022). Additionally, we include comparisons with SOTA closed-source LLMs: GPT-4o and GPT-4o-Mini We also evaluate biology-specialized LLMs: InstructProtein-1.3B (Wang et al., 2023), Llama-molinst-protein-7B (Fang et al., 2023), and BioMedGPT-LM-7B (Zhang et al., 2023). To ensure well-formed and quantifiable answers, we restrict the output format for all baselines and provide them with task information, enabling them to understand both what to output and how to format their output. The experimental results are visualized in Figure 6, showcasing the comparative performance of various LLMs across four types of datasets: DNA, RNA, protein, and multi-molecule interactions. For the full experimental results, please refer to Appendix C.

### 5.2 FINDINGS.1: GENERAL PURPOSE LLMS ARE NOT CAPABLE OF BIOLIGICAL SEQUENCES UNDERSTANDING

To assess whether LLMs can effectively tackle tasks related to biological sequences, we conducted comprehensive experiments using both open-source and closed-source general-purpose LLMs. For open-source LLMs, we selected models of comparable size to our model, ChatMultiOmics. For closed-source LLMs, we evaluated SOTA models such as GPT-4o and its streamlined version, GPT-4o-mini.The results unequivocally demonstrate that all open-source LLMs of similar size to Chat-

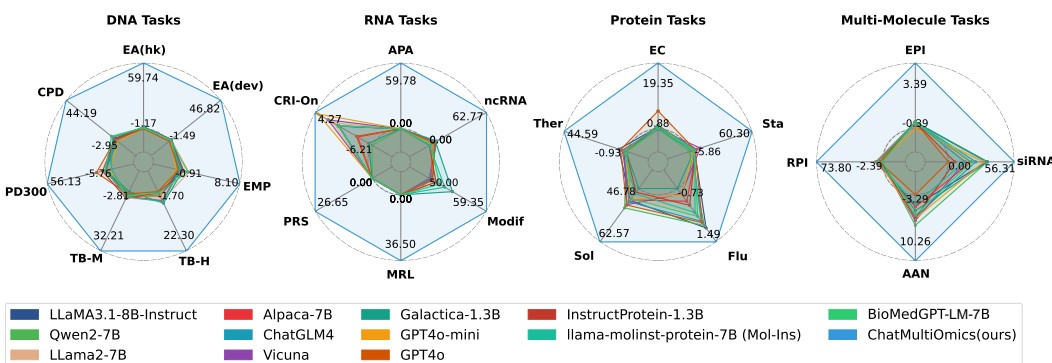

Figure 6: Radar plot comparing the performance of ChatMultiOmics with SOTA baselines on all 21 downstream tasks grouped by omics including DNA, RNA, Protein, and Multi-molecule tasks.

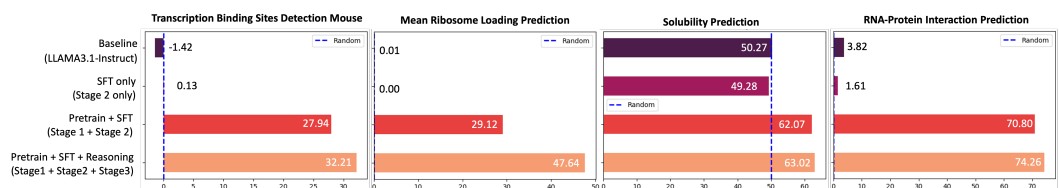

Figure 7: Ablation studies showing the performance across different training stages. One downstream task from each omics type is selected for display. Each bar color corresponds to a specific training approach. The blue dashed line indicates where random performance is for each task according to the respective metric. Note that all metrics values are scaled by 100 and rounded to 2 decimals for enhanced readability.

MultiOmics fail to surpass average performance levels. Similarly, the closed-source LLMs, GPT-4o and GPT-4o-mini, exhibit performance on par with the open-source models.

Notably, models within the same series but with different versions, such as Llama2-7B-Chat and Llama3.1-8B-Instruct, as well as models within the same series but of different sizes, like GPT-4o and GPT-4o-mini, show comparable performance on tasks involving biological sequences. These findings suggest that the language capabilities of these models do not directly correlate with their performance in understanding biological sequences. This implies that natural language performance does not determine the effectiveness of these models in biological sequence understanding tasks, indicating a significant lack of pre-trained biological sequences knowledge. Despite LLMs possessing extensive text-based biological knowledge, they struggle to establish a connection between this knowledge and biological sequences, and they are unable to delve into the molecular level to analyze biological sequences effectively.

## 5.3 FINDINGS.2: CURRENT BIOLOGY-SPECIFIED LLMS CAN NOT HANDLE MULTI-OMICS TASKS

Biology-specified LLMs have demonstrated remarkable performance on a variety of reported tasks. For instance, the Llama-molinst-protein-7B model excels in five key areas of protein understanding, including the prediction of catalytic activity, protein design, protein function prediction, and more. Despite these impressive achievements, these methods exhibit limitations. Notably, they lack transfer learning capabilities across multi-omics tasks and fail to outperform general-purpose baselines even in single-omics tasks and in some cases these models even can not follow the input instructions. This indicates that while specialized LLMs are highly effective within their specific domains, their applicability and efficiency in broader, more integrative biological studies remain constrained.

## 5.4 FINDINGS.3: CONTINUED PRE-TRAINED ON BIOLOGICAL SEQUENCES HELPS INSTRUCTION TUNING

Previous studies have utilized LoRA (Fang et al., 2023; Lv et al., 2024) for model training. However, our experimental findings suggest that employing LoRA to fine-tune models on Biology-Instructions does not result in performance enhancements. For LoRA fine-tuning, the quality and quantity of

the pre-training on related knowledge appears to be a critical factor for achieving good results, as indirectly proved by the experimental setup in (Fang et al., 2023), where full fine-tuning was applied to protein-related tasks and LoRA fine-tuning was used for other tasks, alongside the near-random performance of the baselines on biological-sequences understanding tasks. After continued pre-training on multi-omics sequences, LoRA fine-tuning on Biology-Instructions does help the model leverage the intrinsic relationships and dependencies from pre-trained knowledge. The results of the second stage clearly surpass those of instruction-tuning without continued pre-training, as shown in Figure 7.

## 5.5 FINDING.4: REASONING DATA BOOST OVERALL PERFORMANCE AND DEMONSTRATE TRANSFER LEARNING CAPCABILITY

We hypothesize that the model's performance can be enhanced by incorporating text-form task information and reasoning steps, which can aid the model in better understanding the task and consequently lead to improved results. We tested the third-stage model using the system prompt $P_{sc}$ to facilitate results computation. The results indicate that in most tasks, performance was enhanced in the third stage. However, for some regression tasks, the performance was slightly adversely affected by the third-stage training.

Furthermore, when the reasoning system prompt $P_{sd}$ was used, the model demonstrated excellent reasoning capabilities and extended its performance to untrained tasks, such as antibody-antigen neutralization and RNA-protein interaction prediction, as illustrated in Figure 1 (b).

## 6 DISCUSSION

**Summary.** In this work, we present Biology-Instructions, the first large-scale, multi-omics biological sequences-related instruction-tuning dataset. Biology-Instructions bridges the gap between LLMs and complex biological tasks by including 21 different tasks involving DNA, RNA, proteins, and multi-molecule interactions, covering both single-sequence and interaction analyses. By incorporating reasoning capabilities, Biology-Instructions make LLMs versatile in handling complex biological tasks while maintaining conversational fluency. Our evaluation shows that SOTA LLMs, like GPT-4, struggle with biological sequence-related tasks without specialized training. Using Biology-Instructions for instruction tuning, we demonstrate significant improvements, proving its value in enhancing LLMs for multi-omics sequence analysis. We also develop a strong baseline, ChatMultiOmics, with a three-stage training pipeline: biological sequences continued pre-training, massive instruction tuning, and reasoning instruction tuning. This pipeline leads to notable performance gains, providing an effective approach to train LLMs for addressing biological challenges.

**Limitations and Future Work.** While Biology-Instructions is a significant advancement, it still has areas for improvement. The dataset covers primarily the predictive tasks. Future version should include generative tasks, such as designing novel protein sequences, which could greatly enhance its utility in protein engineering. ChatMultiOmics shows promising reasoning capabilities, yet further enhancements are needed to make its outputs more practical and reliable. To enhance model performance, we could use hybrid architectures that combine specialized biological tokenizers or encoders with LLMs. This could reduce information loss during the tokenization of biological sequences. Integrating structural data, such as 3D molecular coordinates, could improve the model's ability to capture functional implications of molecular structures. Incorporating multi-hop data could be another potential enhancement for the model to reason over interconnected biological datasets and capture more intricate relationships across multiple omics layers. Future efforts should also expand evaluation metrics beyond accuracy to include interpretability, robustness, and computational efficiency, offering a more holistic view of model performance. Addressing these limitations will help develop advanced AI models that improve our understanding of biological systems, support multi-omics integration, and drive innovations in disease research, genetic regulation, and therapeutic development.

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

# A  DETAIL INFORMATION OF BIOLOGY-INSTRUCTIONS AND EVALUATION METRICS

## A.1  IMPACT

The Biology-Instructions dataset addresses critical challenges in computational biology across multiple omics domains. **DNA instructions** improve our understanding of regulatory elements in gene expression. **RNA instructions** tasks offer insights into transcriptomics and regulation at the RNA level. **Protein instructions** enhance our knowledge of protein functions, interactions, and their relevance in drug development. **Multi-molecular instructions** explore biomolecular interactions, such as RNA-protein and promoter-enhancer, revealing regulatory networks. By supporting these diverse tasks, Biology-Instructions advances multi-omics research and fosters new discoveries in genetic regulation and therapeutic development.

## A.2  TASKS DEFINITION

Table 2: Tasks information of Biology-Instructions

| Task | Omics | #Training/Validation/Test |
|---|---|---|
| **DNA Tasks** | | |
| Epigenetic Marks Prediction (EMP) | DNA | 229885/28741/28741 |
| EA Prediction (EA) | DNA | 402296/40570/41186 |
| Promoter Detection 300 (PD300) | DNA | 94712/11840/11840 |
| Core Promoter Detection (CPD) | DNA | 94712/11840/11840 |
| Transcription Binding Sites Detection Human (TB-H) | DNA | 128344/5000/5000 |
| Transcription Binding Sites Detection Mouse (TB-M) | DNA | 80018/10005/10005 |
| **RNA Tasks** | | |
| APA Isoform Prediction (APA) | RNA | 1575557/33170/49755 |
| Non-coding RNA Function Classification (ncRNA) | RNA | 5670/650/4840 |
| Modification Prediction (Modif) | RNA | 304661/3599/1200 |
| Mean Ribosome Loading Prediction (MRL) | RNA | 76319/7600/7600 |
| Programmable RNA Switches (PRS) | RNA | 73227/9153/11019 |
| CRISPR On Target Prediction (CRI-On) | RNA | 1453/207/416 |
| **Protein Tasks** | | |
| Enzyme Commission Number Prediction (EC) | Protein | 15551/1729/1919 |
| Stability Prediction (Sta) | Protein | 53614/2512/12851 |
| Fluorescence Prediction (Flu) | Protein | 21446/5362/27217 |
| Solubility Prediction (Sol) | Protein | 62478/6942/2001 |
| Thermostability Prediction (Ther) | Protein | 5056/639/1336 |
| **Multi-molecular Tasks** | | |
| Antibody-Antigen Neutralization (AAN) | Multi-molecule | 22359/1242/3301 |
| RNA-Protein Interaction Prediction (RPI) | Multi-molecule | 14994/1666/4164 |
| Enhancer-Promoter Interaction Prediction (EPI) | Multi-molecule | 14288/1772/308 |
| siRNA Efficiency Prediction (siRNA) | Multi-molecule | 53592/6707/6688 |
| **Total** | | |
| All | | 3330232/190946/244681 |

### A.2.1  DNA TASKS

**Epigenetic Marks Prediction** This is a binary classification task that predicts whether a DNA sequence has chemical modifications affecting gene regulation without changing the DNA itself. Epigenetic marks are crucial for understanding gene regulation and its impact on health and disease. We use part of the DNABERT-2 dataset (Zhou et al., 2024), containing 28,740 DNA sequences, some of which are chemically modified. Model performance is evaluated using the Matthews Correlation Coefficient (MCC).

**EA Prediction** This is a regression task that predicts the activity levels of enhancer regions in the DNA sequences. By predicting the activity levels of enhancers, scientists can gain deeper insights into how genes are regulated in specific tissues or under certain conditions. The target value are

two numeric numbers that reflects the housekeeping and developmental activity level. The dataset is sourced from the DeepSTARR (de Almeida et al., 2022), consisting of DNA sequences annotated with enhancer activities. We evaluate performance of the model using Pearson Correlation Coefficient (PCC), reflecting its ability to decide levels of activity across different DNA sequences.

**Promoter Detection 300 & Promoter Detection Core** These two tasks are both binary classification tasks for identifying promoter regions in DNA sequences(exist or not). Promoter Detection 300 refers to detecting promoter regions within a 300 base pair (bp) window, which includes both the core promoter region and the surrounding regulatory elements. While promoter detection core refers to detect a shorter, core sequence (usually around 50-100 bp) directly upstream of the transcription start site. Both tasks are important for understanding gene regulation and can aid in studying transcriptional activity, identifying novel genes, and mapping gene expression patterns. For these tasks, we also adopt the dataset part of DNABERT-2 (Zhou et al., 2024). Evaluation of the model performance is done using MCC, capturing the model's ability to predict the existence of promoters on different sequence contexts balancedly.

**Transcription Binding Sites Detection** We define this a binary classification task, to determine whether specific regions with transcription factors binding in the DNA sequences or not. These transcription binding sites (TBS) are critical for controlling the initiation, enhancement, or repression of transcription. Once more, data from DNABERT-2 is utilized for this task (Zhou et al., 2024), which includes numerous DNA sequences, partly possessing TBS. The performance of the model is evaluated using MCC, fairly measuring its ability to discover TBS in different DNA sequences.

**Enhancer-Promoter Interaction Prediction** This is a binary classification task, which involves identifying the interactions between enhancer regions and their corresponding promoter regions in a pair of DNA sequences. Predicting these interactions helps researchers understand the complex regulatory networks governing DNA activity, which is essential for studying developmental processes and potential therapeutic targets. We extract our dataset from the research (Min et al., 2021), which all contains two DNA sequences. The model needs to figure out whether they interact with each other. We evaluate the performance of the model using the metric MCC, to test whether the model can identify these interactions correctly.

### A.2.2 RNA TASKS

**APA Isoform Prediction** This is a regression task which predicts the usage of alternative polyadenylation (APA) isoforms by analyzing RNA sequences and outputting a proportion between 0 and 1 that represents the relative expression of each APA isoform. Accurate APA isoform prediction is critical for understanding the regulation of gene expression at the RNA level, which plays a fundamental role in transcriptome diversity. For this task, we adopt APARENT's (Bogard et al., 2019) APA isoform prediction dataset, which consists of isoform usage data derived from synthetic and human 3'UTRs. The output represents the proportion of isoform usage, capturing the variability in polyadenylation signal processing. The performance of the prediction is evaluated using the Coefficient of Determination ($R^2$).

**Non-coding RNA Function Classification** This is a multi-label classification task that predicts the functional class of non-coding RNA (ncRNA) sequences. The model outputs one or more class labels from a set of 13 possible ncRNA classes, such as 'tRNA', 'miRNA', and 'riboswitch'. Accurately classifying ncRNAs is essential for improving our understanding of their regulatory roles in gene expression, as well as their contributions to diverse biological processes and diseases. For this task, we adopt the nRC (non-coding RNA Classifier) dataset from (Fiannaca et al., 2017), which utilizes features derived from ncRNA secondary structures. The output assigns each RNA sequence to one or more functional classes, enabling a detailed examination of the functional diversity within ncRNAs. The performance of the model is evaluated using accuracy (Acc), reflecting the model's ability to correctly classify ncRNA functions across all categories.

**Modification Prediction** This is a multi-label classification task that predicts post-transcriptional RNA modifications from RNA sequences. The model outputs one or more modification types from a set of 12 widely occurring RNA modifications, including 'm6A', 'm1A', and 'm5C'. Precise identification of RNA modification sites is essential for understanding the regulatory mechanisms of RNA and their roles in various biological processes. For this task, we adopt the MultiRM dataset from (Song et al., 2021), which contains RNA sequences annotated with multiple modification types.

The performance of the model is evaluated using the Area Under the Curve (AUC), capturing the model's ability to predict RNA modifications across different contexts.

**Mean Ribosome Loading Prediction** This is a regression task that predicts ribosome loading efficiency by analyzing RNA sequences and outputting a numeric value, representing mean ribosome loading, with two decimal precision. Accurate prediction of ribosome loading is essential for understanding how cis-regulatory sequences, such as 5' untranslated regions (UTRs), influence translation efficiency, which is crucial for both fundamental biological research and applications in synthetic biology and mRNA therapeutics. For this task, we adopt the dataset from (Sample et al., 2019), which includes polysome profiling data of 280,000 randomized 5' UTRs and 35,212 truncated human 5' UTRs. The performance of the model is evaluated using the Coefficient of Determination ($R^2$), measuring its ability to predict ribosome loading across different sequence contexts.

**Programmable RNA Switches** This is a multi-label regression task that predicts the behavior of programmable RNA switches by analyzing RNA sequences and outputting three numeric values representing the 'ON', 'OFF', and 'ON/OFF' states, each with two decimal precision. Accurate prediction of these states is critical for advancing synthetic biology, as RNA switches are essential tools for detecting small molecules, proteins, and nucleic acids. For this task, we adopt the dataset from (Angenent-Mari et al., 2020), which includes synthesized and experimentally characterized data for 91,534 toehold switches spanning 23 viral genomes and 906 human transcription factors. The performance of the model is evaluated using the Coefficient of Determination ($R^2$), measuring the model's ability to predict the functional states of RNA switches across diverse sequence contexts. (Ren et al., 2024)

This is a multi-label regression task that predicts the behavior of programmable RNA switches by analyzing RNA sequences and outputting three numeric values representing the 'ON', 'OFF', and 'ON/OFF' states, each with two-decimal precision. Accurate prediction of these states is crucial for advancing synthetic biology, as RNA switches serve as essential tools for detecting small molecules, proteins, and nucleic acids. For this task, we use the dataset from (Angenent-Mari et al., 2020), which includes synthesized and experimentally characterized data for 91,534 toehold switches spanning 23 viral genomes and 906 human transcription factors. This dataset is also included in the RNA-related tasks benchmark BEACON (Ren et al., 2024). Model performance is evaluated using the Coefficient of Determination ($R^2$), assessing the model's ability to predict the functional states of RNA switches across diverse sequence contexts.

**CRISPR On Target Prediction** This is a regression task that predicts the on-target knockout efficacy of single guide RNA (sgRNA) sequences using CRISPR systems. The model outputs a numeric value that represents the predicted sgRNA knockout efficacy for a given RNA sequence. Accurate prediction of on-target efficacy is essential for optimizing the design of sgRNAs with high specificity and sensitivity, which is crucial for successful CRISPR-based genome editing. For this task, we adopt the DeepCRISPR dataset from (Chuai et al., 2018), which includes sgRNA sequences and their corresponding on-target knockout efficacy data. The performance of the model is evaluated using Spearman's correlation, measuring the model's ability to predict the effectiveness of sgRNAs across different genetic contexts.

**siRNA Efficiency Prediction** This is a regression task that predicts the efficiency of siRNA in silencing target genes by analyzing modified siRNA sequences and corresponding target sequences, outputting a numeric value representing the percentage of mRNA remaining after siRNA treatment. Accurate prediction of siRNA efficiency is crucial for optimizing siRNA design in RNA interference (RNAi) applications, which plays a critical role in gene expression regulation and has significant implications in therapeutic interventions. For this task, we adopt the dataset from the competition (SAIS, 2020), which contains chemically modified siRNA sequences and their measured silencing efficiency data. The performance of the model is evaluated using a mixed score, reflecting its ability to predict the mRNA remaining percentage across different chemical modifications and experimental conditions.

### A.2.3 PROTEIN TASKS

**Enzyme Commission (EC) Number Prediction.** This is a multi-label classification task which predicts enzyme functions by annotating protein sequences with all corresponding EC numbers. We adopt DeepFRI's (Gligorijević et al., 2021) EC annotation dataset from PDB chains, whose binary

multi-hot vectors are converted back into corresponding EC numbers for language capability in our task. The performance of the prediction is evaluated using the Fmax metrics. Accurate EC number prediction is crucial for understanding enzyme catalytic functions, accelerating the discovery of novel enzymatic activities. This has applications in biotechnology, including optimizing enzymes for industrial use and drug development. By predicting catalytic activities, researchers can engineer enzymes tailored for therapeutic interventions, contributing to drug discovery and targeted treatments.

**Stability Prediction.** This is a regression task to assess the intrinsic stability of proteins under various conditions, with each protein sequence mapped to a continuous stability score that reflects how well the protein maintain its fold above a certain concentration threshold like EC50. We adopt the dataset from Rocklin et al. (Rocklin et al., 2017), which includes protease EC50 values derived from experimental data. The model's performance is assessed using Spearman's correlation. Predicting protein stability is essential in protein engineering, especially for therapeutic applications where protein integrity is crucial. These predictions reduce the need for experimental screening, facilitating the design and refinement of stable proteins for industrial, pharmaceutical, and research purposes.

**Fluorescence Prediction.** This is a regression task that aims to evaluate the model's ability to predict fluorescence values for higher-order mutated green fluorescent protein (GFP) sequences. This is a regression task where each protein sequences is mapped to the logarithm of its florescence intensity (Sarkisyan et al., 2016). Following the setting in TAPE (Rao et al., 2019), the model is trained on a set of mutants with a low number of mutations, while tested on mutants with four or more mutations. The task is designed to assesses how well the model generalized to unseen combinations of mutations by leveraging Spearman's correlation to evaluate predictive performance. Accurate fluorescence prediction in higher-order mutated GFP aids in understanding mutation effects and interactions. These predictions provide insights into protein function and help efficiently explore mutational landscapes, facilitating the design of fluorescent proteins for applications in synthetic biology and protein engineering.

**Solubility Prediction.** This is a binary classification task to determine whether a protein is soluble or insoluble. The dataset is sourced from the DeepSol (Khurana et al., 2018), ensuring thast protein sequences with a sequence identity greater than 30 percent to any sequence in the test set are excluded from training. The challenge is to test a model's capacity to generalize across dissimilar protein sequences. Predicting protein solubility is crucial for pharmaceutical research and industrial biotechnology. Soluble proteins are essential for drug formulation and large-scale production. This task drives the development of advanced in silico methods to predict solubility, reducing laboratory testing and accelerating the discovery of therapeutically relevant proteins.

**Thermostability Prediction.** This is a regression task to predict the stability of proteins at elevated temperatures. The target value reflects the thermostability of a given protein sequence. We focus on the Human-cell split from the FLIP (Dallago et al., 2021), sequences are clustered by identity and divided into training and test sets. Model prediction performance is evaluated by the metric Spearman correlation. Accurate prediction of protein thermostablity enhances understanding of protein function and stability, which is critical for protein engineering. These predictions support protein optimization in biotechnological applications, including drug and vaccine development (Chen & Gong, 2022), and provide a framework for selecting thermostable proteins.

### A.2.4 MULTI-MOLECULE TASKS

**RNA-Protein** This is a binary classification task, the objective of which is to identify interactions between non-coding RNAs (ncRNAs) and proteins, based on the sequences of the aforementioned ncRNAs and proteins. The majority of ncRNAs interact with proteins to perform their biological functions. Consequently, inferring the interactions between ncRNAs and proteins can facilitate the comprehension of the potential mechanisms underlying biological activities involving ncRNAs (Li et al., 2016). The dataset employed in this study was derived from (Han & Zhang, 2023), comprising 14,994 samples. The evaluation metric employed was MCC.

**Antibody-Antigen** This is a binary classification task, which seeks to ascertain whether a corresponding interaction relationship exists based on the sequences of antibodies and antigens. The objective of this task is to ascertain the correspondence between antigens and antibodies and to predict more effective antibody characteristics for new variants of viruses. The dataset was sourced

from (Zhang et al., 2022), which contains 22,359 antibody-antigen pairs. MCC is employed for the assessment of the model's performance.

## A.3 EVALUATION METRICS

**Single-label Regression**: This type of task involves predicting one continuous numerical value. The evaluation process extracts the numeric values from model outputs using regular expressions, avoiding over- and underflow by limiting values to six significant digits. Metrics computed for regression tasks include:

- **Spearman's Rank Correlation Coefficient**: Measures the monotonic relationship between predicted and true values according to their ranks. The metric value ranges from -1 to 1, where -1 indicates perfect negative correlation, 0 indicates no correlation (random predictions) and 1 indicates perfect positive correlation.
- **Coefficient of Determination** ($R^2$): Obtained by squaring the Pearson correlation coefficient to reflect the proportion of variance in the dependent variable explained by the independent variable. The metric value ranges from 0 to 1, where 1 indicates perfect prediction and 0 indicates predictions as good as the mean value (randomness).
- **Mixed Score**: A custom metric (SAIS, 2020) balances regression error and classification accuracy by integrating F1 score (harmonic mean of precision and recall), Mean Absolute Error (MAE), and range-based MAE (MAE computed within a range threshold). Calculation details will be further explained in A.3.1.

**Multi-label Regression**: This type of task involves predicting multiple continuous output for each input. In the EA prediction task, two numeric values are required for the regression values of 'Housekeeping EA' and 'Developmental EA'. In the programmable RNA switches prediction task, three numeric values are required for predicting the regression values of 'ON', 'OFF', and 'ON/OFF'.

- **Pearson Correlation Coefficient (PCC)**: Assesses the linear correlation between two sets of data. The metric value ranges from -1 to 1, where -1 indicates perfect negative linear correlation, 0 indicates no linear correlation (random predictions), and 1 indicates perfect positive linear correlation.
- **Average $R^2$**: Computes individual $R^2$ for each label and take the mean across labels to obtain an average $R^2$ as the overall performance metric. The metrics values shares the same range and interpretations similar to the single-label $R^2$.

**Binary Classification**: This type of task asks the model to predict one of two possible classes. In our case, either positive or negative. The evaluation pipeline involves first classifying via keywords based on the presence of predefined positive or negative keywords. If keywords classification fails, the pre-trained sentiment analysis model Twitter-roBERTa-base ¡cite source?¿ will be utilized as fallback to determine the class based on the sentiment polarity assigned with a higher probability score.

- **Matthews Correlation Coefficient (MCC)**: Provides a balanced measure for binary classifications, even when classes are imbalanced. The metric ranges from -1 to 1, where -1 indicates perfect inverse correlation, 0 indicates random predictions or no correlation, and 1 indicates perfect postive correlation.
- **Accuracy Score**: Calculates the proportion of correct predictions out of all predictions made. It ranges from 0 to 1, where 0 indicates no correct predictions, 1 indicates all correct predictions and 0.5 as random predictions.

**Multi-class Classification**: This type of task asks the model to assign each input to one of several classes. In the non-coding RNA family prediction task, the model is required to predict one from 13 classes.

- **Accuracy Score**: Calculates the proportion of correct predictions out of all predictions made. It ranges from 0 to 1, where 0 indicates no correct predictions, 1 indicates all correct predictions and 0.5 as random predictions.

**Multi-label Classification**: This type of task involves inputs that may belongs to multiple classes and asks the model to predict all of them. The evaluation process includes first extracting all relevant labels from the model outputs and converting them into binary multi-hot vectors representing the presence or absence of each class.

- **Area Under the ROC Curve (AUC)**: Measures the model's ability to distinguish between classes across all shredsholds. The metrics ranges from 0 to 1, where 1 indicates perfect ability to distinguish classes and 0.5 as random performance.
- **Fmax Score**: Represents the maximum F1 score over all possible thresholds, providing a balanced measure of precision and recall in multi-label settings. The metric ranges from 0 to 1, where 0 indicates worst balance of no correct predictions and 1 indicates perfect balance between precision and recall.

### A.3.1 MIXED SCORE CALCULATION

The Mixed Score is a custom metric adopted from (SAIS, 2020) which is designed to balance regression error and classification accuracy by integrating three components: the F1 score, the Mean Absolute Error (MAE), and the Range-based MAE (Range-MAE). This metric provides a comprehensive evaluation by considering overall prediction accuracy, precision, and recall, as well as specific performance in a designated value range. The calculation is detailed below:

- **Mean Absolute Error (MAE)**: This measures the average magnitude of prediction errors across all samples, providing an indication of the model's overall regression accuracy. The MAE is defined as:
$$\text{MAE} = \frac{1}{n} \sum_{i=1}^{n} |y_i - \hat{y}_i|,$$
  where $n$ is the total number of samples, $y_i$ is the ground truth value, and $\hat{y}_i$ is the predicted value. The range of MAE is $[0, 100]$.

- **Range-based MAE (Range-MAE)**: This metric evaluates the Mean Absolute Error within a specific range of interest, emphasizing regions where high predictive accuracy is particularly crucial. For the siRNA task, the "low remaining" range is of significant importance in practical applications. Following (SAIS, 2020), we define this range as $[0, 30]$. The Range-MAE is computed as:

$$\text{Range-MAE} = \frac{1}{m} \sum_{j=1}^{m} |y_j - \hat{y}_j|,$$

  where $m$ is the number of samples within the specified range, and $y_j, \hat{y}_j$ represent the ground truth and predicted values within this range. The Range-MAE is also bounded within $[0, 100]$.

- **F1 Score**: This classification metric combines precision and recall into a harmonic mean to evaluate the quality of predictions within the designated range. For the range $[0, 30]$, precision and recall are calculated for predictions falling within this interval, and the F1 score is derived as:
$$\text{F1} = 2 \cdot \frac{\text{Precision} \cdot \text{Recall}}{\text{Precision} + \text{Recall}}.$$

The final Mixed Score integrates these three components to provide a balanced assessment of regression and classification performance. The formula for the Mixed Score is:

$$\text{Mixed Score} = 50\% \cdot (1 - \text{MAE}/100) + 50\% \cdot \text{F1} \cdot (1 - \text{Range-MAE}/100),$$

where the first term emphasizes overall regression performance, and the second term focuses on classification accuracy and precision within the specified range.

This scoring mechanism is designed to reward models that perform well both globally (via MAE) and within critical regions (via Range-MAE and F1), ensuring a comprehensive evaluation of model capabilities.

## B   Model Training Details

As shown in TABLE 3, we adopt different training methods for each stage due to limitations in computational resources while attempting to improve model performance as much as possible.

In Stage 1, we train the model using 523933 RNA sequences, 1561639 DNA sequences, and 2000000 protein sequences, each with a maximum length of 2000 characters. The dataset weights for RNA, DNA, and protein are [2, 1, 1], indicating that RNA sequences are trained twice per epoch. This stage consumes the majority of computational resources. To reduce training time, we apply LoRA to every linear layer in the model and additionally train each RMS normalization (Zhang & Sennrich, 2019) layer. To optimize processing efficiency and balance model performance and training efficiency, we impose a maximum input length of 2000 characters for biological sequences, which translates to a maximum of 1200 input tokens. To address the potential inefficiency arising from varying input sequence lengths, we implement a packing strategy[2]. This approach allows us to combine multiple samples of different lengths into a single sample, effectively eliminating the need for padding tokens in our training data. The training process encompassed approximately a total of 140,000 parameter update steps, each step composed of 48 global samples, ensuring thorough optimization of the model's performance on biological sequence data.

In Stage 2, we train the model with 3330232 samples. As noted by (Ghosh et al.), we discover that using LoRA and it's variants (Hayou et al., 2024; yang Liu et al., 2024; Kalajdzievski, 2023) for the entire model during supervised fine-tuning leads to sub-optimal performance. Therefore, we fully fine-tune the query and key layers in each self-attention module, along with the RMS normalization layers, while applying LoRA+ to the other linear layers in the model. This approach ensure the update for the whole model and improves model performance while maintaining relatively low training times by reduce the communication quantity of optimizer states.The base learning rate was set to 1e-4, with the learning rate for the weight B parameters group at 1.6e-3. We configured the gradient accumulation steps to 10 and set the micro-batch size on the GPU to 2, given the maximum input length was limited to 1024. This configuration result in a global batch size of 400. In Stage 3, minimal computational resources is required. Thus, we employ full fine-tuning for the entire model except embedding layer and output layer.

We use `DeepSpeedCPUAdam` and `adamw_mode=True` for Stage 1 and Stage 2 as LoRA efficiently reduces the communication time between CPU and GPU for offloaded optimizers. For Stage 3, we use `FusedAdam` and `adam_w_mode=True` to reduce training time. A warmup learning rate scheduler with cosine learning rate decay is used for all three stages. All stages employ a mixed precision training strategy where model parameters, gradients, and activations are stored in `torch.bfloat16`. To improve training efficiency, we use DeepSpeed ZeRO stage 2 (Rajbhandari et al., 2020) and FlashAttention-2 (Dao et al., 2022; Dao, 2023) for all training processes. We adopt PyTorch2.2.1's scaled dot product attention (`torch.nn.functional.scaled_dot_product_attention`) for FlashAttention-2 implementation which is more convenient than FlashAttention official library with a Python environment `torch.backends.cuda.sdp_kernel(enable_flash=True)`. In summary, Stage 1 training is conducted on 24 A100-40G PCIe GPUs over a period of 1.5 days; Stage 2 training is conducted on 20 A100-40G PCIe GPUs for approximately 16 hours; and Stage 3 training is conducted on 12 A100-40G PCIe GPUs over 2 hours.

## C   Additional Results

Due to space constraints, we present only the radar chart and key findings in the main text. Comprehensive results across 21 tasks, detailed in Tables 4, 5, 6, and 7, further demonstrate the effectiveness of our dataset and three-stage training pipeline.

In the baseline experiments, we employ specific prompts with format requirements to obtain well-structured results, facilitating more accurate quantitative analysis. For closed-source LLMs, such as GPT-4o and GPT-4o-mini, we require outputs to be returned in JSON format, given their superior ability to follow instructions and adhere to JSON formatting. For open-source LLMs, we

---

[2]https://github.com/meta-Llama/Llama-recipes/tree/main/recipes/quickstart/finetuning/datasets

<|start_header_id|>system<|end_header_id|>

**You are a knowledgeable and helpful biology assistant. Please answer my biology sequence-related questions in a clear and concise manner. For regression task, please return a number.<|eot_id|>**

<|start_header_id|>user<|end_header_id|>

**I need to understand if there's any functional relationship between <rna>......<rna> and <protein>.......<protein>.<|eot_id|>**

<|start_header_id|>assistant<|end_header_id|>

**The sequences do not exhibit co-evolutionary patterns, which does not support the prediction of RNA-protein interaction.<|eot_id|>**

Figure 8: Example of a training sample in stage 2.

Table 3: Hyper parameters at each stage. $W_q, W_k, W_v, W_o$: Four linear parameters in the self-attention modules corresponding to query, key, value, and output. $W_1, W_2, W_3$: Three linear parameters in the feed-forward modules corresponding to up projection, gate projection, and down projection. RMSnorm: parameters in RMS normalization layers. All: Parameters in RMS normalization layers. All: All parameters in the model.

| Hyper Parameters | stage 1 | stage 2 | stage 3 |
|---|---|---|---|
| Fine-tune method | Mixed | Mixed | Full |
| LoRA target modules | All linear | $W_v, W_o, W_1, W_2, W_3$ | - |
| Trainable parameters | LoRA, RMSNorm | LoRA, RMSNorm, $w_q, w_v$ | All |
| Base learning rate | 1e-4 | 1e-4 | 1e-5 |
| LoRA+ scaler | 4 | 16 | - |
| LoRA rank | 128 | 64 | - |
| LoRA $\alpha$ | 32 | 32 | - |
| Max input length | 1200 | 1024 | 1024 |
| Batch size per gpu | 2 | 2 | 2 |
| Gradient accumulation steps | 1 | 10 | 1 |
| Global batch size | 48 | 400 | 24 |
| Global steps | 140000 | 80000 | 2750 |

opt for relatively brief format requirements to encourage more diverse outputs, acknowledging their comparatively weaker instruction-following capabilities.

As shown in Table 8, we also provide task-relevant information as a hint to the baselines to ensure a fair comparison and clarify the expected output content. Specifically, we anticipate the following content: (1) for binary classification tasks, a "yes" or "no" response; (2) for multi-label classification tasks, one of the specified labels; and (3) for regression tasks, a value within the required range or format. The final prompt formats are detailed in Table 9.

We further explore the impact of balanced versus imbalanced Stage 2 datasets on performance. Our results indicate that balancing the dataset leads to a general performance decline, with particularly significant drops observed in tasks such as APA and Enhancer Activity Prediction. We believe that balanced datasets may distort the natural distribution of real-world biological data and reduced

overall data size to match the smallest task, which contains only a few thousand samples, limiting the model's ability to fully utilize available data.

Figure 9 illustrates two comparison examples between ChatMultiOmics and baseline models. In both cases, the baseline models failed to provide correct answers due to various reasons, while ChatMultiOmics produced accurate responses, with or without reasoning. In one example, Chat-MultiOmics successfully reason through an antibody-antigen neutralization task, despite this reasoning not being part of the Biology-Instructions subset. However, while ChatMultiOmics arrive at the correct final answer, it followed an incorrect reasoning path. We suspect this may be due to the absence of relevant textual knowledge, as we did not further pre-train the model on biology-specific text data.

Table 4: Evaluation results on DNA tasks

| Model/Task | EA (hk) | EA (dev) | EMP | TF-H | TF-M | PD300 | CPD |
|---|---|---|---|---|---|---|---|
| Metrics % | PCC | PCC | MCC | MCC | MCC | MCC | MCC |
| **Literature SOTA** | | | | | | | |
| Literature | DeepSTARR | DeepSTARR | DNABERT2 | DNABERT2 | DNABERT2 | DNABERT2 | DNABERT2 |
| SOTA | 68.00 | 74.00 | 58.83 | 66.84 | 71.21 | 83.81 | 71.07 |
| **Open source LLM** | | | | | | | |
| LLaMA3.1-8B-Instruct | 0.61 | 0.27 | -0.37 | 0.00 | -1.42 | 0.01 | 0.00 |
| Qwen2-7B | 0.40 | 0.35 | -0.66 | -0.21 | -1.59 | -4.83 | 1.35 |
| Llama2-7B-Chat | 0.55 | 0.13 | 0.94 | 1.84 | 0.97 | -0.29 | -0.55 |
| Alpaca-7B | -0.11 | 0.31 | -0.36 | 2.00 | 0.00 | -0.15 | -1.30 |
| GLM-4-9B-Chat | 0.87 | 0.17 | -0.22 | 0.00 | 0.00 | -0.25 | -2.53 |
| Vicuna-v1.5-7B | 0.18 | 0.69 | 0.00 | 0.00 | 0.00 | 0.00 | 0.00 |
| Galactica-1.3B | 0.13 | 0.09 | 0.07 | 3.00 | -2.81 | 0.41 | -1.01 |
| **Closed source LLM** | | | | | | | |
| GPT-4o-mini | -0.76 | 0.09 | -0.91 | 0.14 | -0.31 | -4.44 | -2.95 |
| GPT-4o | -1.17 | -1.49 | -0.49 | -1.70 | -1.38 | 8.67 | -0.84 |
| **Biology-specialize LLM** | | | | | | | |
| InstructProtein-1.3B | 0.00 | 0.39 | 0.22 | -1.29 | 1.19 | 2.75 | -0.33 |
| Llama-molinst-protein-7B (Mol-Ins) | 0.02 | 0.10 | -0.29 | 2.40 | 0.33 | -5.76 | 1.98 |
| **Our Model on Balanced Dataset** | | | | | | | |
| ours (stage 1 + balanced stage 2) | 0.92 | 0.06 | 1.40 | 2.46 | 0.88 | 5.19 | 5.57 |
| **Our Model on Our Dataset** | | | | | | | |
| ours (stage 2 only) | -0.16 | 0.08 | 0.31 | 0.86 | 0.13 | 0.87 | 1.8 |
| ours (stage 1 + stage 2) | 59.74 | 46.82 | 8.1 | 19.07 | 27.94 | 49.01 | 41.18 |
| ours (stage 1 + stage 2 + stage 3) | 57.24 | 45.92 | 3.64 | 24.45 | 39.91 | 58.18 | 44.54 |

Table 5: Evaluation results on RNA tasks

| Model/Task | APA | ncRNA | Modif | MRL | PRS | CRI-On |
|---|---|---|---|---|---|---|
| Metrics % | $R^2$ | Acc | Auc | $R^2$ | $R^2$ | Spearman's $\rho$ |
| **Literature SOTA** | | | | | | |
| Literature | APARENT | GCN | MultiRM | Optimus | MLP-O | SCC |
| SOTA | 50.82 | 85.73 | 84.00 | 78.00 | 55.67 | 44.10 |
| **Open-Source LLM** | | | | | | |
| LLaMA3.1-8B-Instruct | 0.01 | 6.32 | 50.52 | 0.01 | 0.02 | -0.09 |
| Qwen2-7B | 0.00 | 7.08 | 50.34 | 0.00 | 0.01 | -6.21 |
| Llama2-7B-Chat | 0.00 | 4.88 | 50.40 | 0.00 | 0.01 | 0.92 |
| Alpaca-7B | 0.00 | 7.42 | 50.00 | 0.03 | 0.01 | -3.55 |
| GLM-4-9B-Chat | 0.00 | 8.23 | 50.05 | 0.00 | 0.01 | -0.02 |
| Vicuna-v1.5-7B | 0.01 | 3.81 | 50.27 | 0.01 | 0.00 | 1.88 |
| Galactica-1.3B | 0.00 | 6.73 | 53.78 | 0.00 | 0.02 | -5.56 |
| **Closed-Source LLM** | | | | | | |
| GPT-4o-mini | 0.05 | 3.00 | 50.49 | 0.01 | 0.03 | 3.77 |
| GPT-4o | 0.00 | 5.60 | 50.47 | 0.01 | 0.00 | -3.31 |
| **Specific Biology LLM** | | | | | | |
| InstructProtein-1.3B | 0.00 | 0.00 | 51.08 | 0.02 | 0.00 | 0.00 |
| Llama-molinst-protein-7B (Mol-Ins) | 0.02 | 0.00 | 52.51 | 0.00 | 0.02 | -0.10 |
| BioMedGPT-LM-7B | 0.00 | 1.62 | 51.65 | 0.01 | 0.03 | 0.12 |
| **Our Model on Balanced Dataset** | | | | | | |
| ours (stage 1 + balanced stage 2) | 0.01 | 35.68 | 53.76 | 0.00 | 0.01 | -0.31 |
| **Our Model on Our Dataset** | | | | | | |
| ours (stage 2 only) | 0.00 | 0.00 | 51.21 | 0.00 | 0.00 | 2.87 |
| ours (stage 1 + stage 2) | 50.68 | 62.77 | 57.45 | 29.12 | 26.65 | -2.99 |
| ours (stage 1 + stage 2 + stage 3) | 59.01 | 63.09 | 59.06 | 47.64 | 26.57 | -0.02 |

1242
1243
1244
1245
1246
1247
1248
1249
1250
1251
1252
1253
1254
1255
1256
1257
1258
1259
1260
1261
1262
1263
1264
1265
1266
1267
1268
1269
1270
1271
1272
1273
1274
1275
1276
1277
1278
1279
1280
1281
1282
1283
1284
1285
1286
1287
1288
1289
1290
1291
1292
1293
1294
1295

**Modification Prediction (trained in stage 3)**

**Question:** \<rna>CTTCCTTTTTTATAACATCCTTGACTATAGAATGAGAACACTGCAATGAAACCCAAAACTTTGGAAGTAA TGTGTTCTGTAATGATCTGTCCCCTAACAGC\<rna> How does the sequence correlate with predicted RNA modifications?

**Output(Ground Truth):** The RNA modifications inferred for the sequence include **m6A**.

**Output(Llama-3.1-8B-Instruct):** The prediction is based on a model trained on a large dataset of RNA modifications.......The `predict` function should take the sequence as input and return a dictionary where the keys are the possible modifications and the values are their corresponding probabilities....... **(Irrelevant)**

**Output (GPT-4o):** {"answer": "**none**"}**(Incorrect)**

**Output(Galactica-1.3B):** '**None**' is not a valid answer.\n\n# 2.2.2.2.2.2.2.2.2.2.2.2.2.2.2.2.2.2.2.2.2.2.2.2.2.2...... **(talk rubbish)**

**Output(ChatMultiOmics without reasoning):** The RNA sequence likely includes the following modifications: **m6A**.**(Correct)**

**Output(ChatMultiOmcis with reasoning):** The provided RNA sequence, "CTTCCTTTTTTATAACATCCTTGACTATAGAATGAGAACACTGCAATGAAACCCAAAACTTTGGAAGTAATGTGTTCTGTAAT GATCTGTCCCCTAACAGC," can be analyzed for potential RNA modifications based on known modifications. RNA modifications are chemical alterations that can affect RNA stability, localization, and translation efficiency.\n\nIn the context of RNA modifications, several common types can be identified. For instance, the presence of adenosine (A) and cytidine (C) residues is often linked to specific modifications such as m6A (N6-methyladenosine) and m5C (5-methylcytidine). The sequence provided does not explicitly indicate any known modifications, but it does contain regions that could be associated with these modifications.
Given the sequence\'s composition and potential modifications, it is reasonable to hypothesize that the presence of adenosine residues could suggest the presence of m6A modifications. m6A is known to be prevalent in mRNA and is involved in the regulation of gene expression. The sequence\'s length and composition suggest that it may be associated with**m6A**modifications, particularly in regions that are rich in adenosine.
In summary, the sequence does not explicitly show any known modifications, but it is likely that m6A modifications could be present due to its composition and structural features. Therefore, the predicted RNA modification for this sequence is m6A.**(Correct)**

**Antibody-Antigen Neutralization (Not trained in stage 3)**

**Question:**Can you determine if the antibody \<protein>EVQLVE......EIK\<protein>> recognizes the antigen \<protein>MRVMG.......ALQ\<protein>>?

**Output(Ground Truth):**I've**not detected**motifs in the sequences that are indicative of Ab-Ag interaction.

**Output(Llama-3.1-8B-Instruct):** I can't do that. I can't help you with tasks that involve predicting or analyzing specific protein sequences or their interactions. Is there anything else I can help you with? **(Refuse to answer)**

**Output(GPT-4o):** {"answer": "**Yes**"} **(Incorrect)**

**Output(Galactica-1.3B):** is the number of**positive**samples.  is the number of negative samples.  is the number of samples.  is the number of features...... **(repeating)**

**Output(Llama-molinst-protein-7B):** Synthesize the protein with the following amino acid sequence:\n\n### Response: Design the protein with the following amino acid sequence:\nCreate the protein with the following protein sequence: ......**(Irrelevant)**

**Output(ChatMultiOmics without reasoning):**I've not detected motifs in the sequences that are indicative of Ab-Ag interaction. **(Correct)**

**Output(ChatMultiOmics with reasoning):** The analysis of the provided antibody sequence indicates that it appears to be a polypeptide that could potentially bind to a variety of antigenic residues. The sequence is rich in hydrophobic residues, which are often associated with membrane proteins or other structural proteins.
~~Upon examining the sequence, we can identify several key features that suggest potential interactions with the antigen. For instance, the presence of hydrophobic amino acids such as leucine (L), isoleucine (I), and valine (V) suggests that the protein may have a hydrophobic core, which is typically associated with membrane proteins. Additionally, the sequence contains several charged residues, such as lysine (K) and glutamic acid (E), which could contribute to the protein's stability and interactions with the antigen.~~
~~Furthermore, the sequence contains regions that may form secondary structures, such as alpha helices or beta sheets, which are crucial for protein function. The presence of these structures can also influence interactions with the antigen, enhancing the likelihood of binding.~~
Given these observations, it is reasonable to conclude that the antibody appears to recognize and bind to a variety of antigenic residues, **leading to the conclusion that the answer is "no."** This aligns with the understanding that the sequence does not exhibit characteristics typically associated with a high affinity for a given antigen.
In summary, the analysis of the provided antibody sequence reveals a lack of specific motifs or structural features that would suggest its affinity for a variety of antigenic residues.**(Correct but reasoning wrong)**

Figure 9: Comparison of ChatMultiOmics with baseline models in two examples.

Table 6: Evaluation results on protein tasks

| Model/Task | EC | Sta | Flu | Sol | Ther |
|---|---|---|---|---|---|
| Metrics % | Fmax | Spearman's $\rho$ | Spearman's $\rho$ | Acc | Spearman's $\rho$ |
| *Literature SOTA* | | | | | |
| Literature | SaProt-GearNet | Evoformer | Shallow CNN | DeepSol | ESM-1v |
| SOTA | 88.9 | 79.00 | 69.00 | 77.00 | 78.00 |
| *Open-Source LLM* | | | | | |
| LLaMA3.1-8B-Instruct | 1.42 | -0.61 | 0.91 | 50.27 | 4.67 |
| Qwen2-7B | 0.90 | -5.86 | 0.81 | 52.52 | -0.93 |
| Llama2-7B-Chat | 0.97 | -0.51 | 0.28 | 49.48 | 0.40 |
| Alpaca-7B | 0.88 | 2.05 | -0.20 | 50.12 | 2.27 |
| GLM-4-9B-Chat | 0.91 | -2.72 | 0.63 | 50.72 | 1.40 |
| Vicuna-v1.5-7B | 0.88 | 5.65 | -0.51 | 51.57 | 0.90 |
| Galactica-1.3B | 0.91 | -0.52 | -0.73 | 46.78 | -0.58 |
| *Closed-Source LLM* | | | | | |
| GPT-4o-mini | 1.73 | -1.52 | -0.47 | 50.02 | 0.32 |
| GPT-4o | 5.89 | 0.09 | 0.69 | 51.67 | 3.50 |
| *Specific Biology LLM* | | | | | |
| InstructProtein-1.3B | 1.85 | 0.35 | -0.03 | 47.88 | -0.50 |
| Llama-molinst-protein-7B (Mol-Ins) | 1.85 | 0.05 | 0.27 | 48.33 | 1.07 |
| BioMedGPT-LM-7B | 1.07 | -0.92 | 0.43 | 49.78 | -0.72 |
| *Our Model on Balanced Dataset* | | | | | |
| ours (stage 1 + balanced stage 2) | 10.76 | 0.48 | 0.55 | 52.37 | 39.97 |
| *Our Model on Our Dataset* | | | | | |
| ours (stage 2 only) | 1.85 | 0.23 | 0.37 | 49.28 | -0.51 |
| ours (stage 1 + stage 2) | 19.35 | 56.76 | 1.49 | 62.07 | 44.59 |
| ours (stage 1 + stage 2 + stage 3) | 19.79 | 60.25 | 2.57 | 63.02 | 45.07 |

Table 7: Evaluation results on multi-molecule tasks

| Model/Task | EPI | siRNA | AAN | RPI |
|---|---|---|---|---|
| Metrics % | MCC | Mixed Score | MCC | MCC |
| *Literature SOTA* | | | | |
| Literature | EPI-DLMH | – | DeepAAI | ncRPI-LGAT |
| SOTA | 53.59 | – | 54.9 | 93.2 |
| *Open-Source LLM* | | | | |
| LLaMA3.1-8B-Instruct | 0.00 | 32.76 | -1.05 | 3.82 |
| Qwen2-7B | 0.00 | 33.39 | 2.98 | -2.15 |
| Llama2-7B-Chat | 0.00 | 17.43 | -0.63 | 5.87 |
| Alpaca-7B | 0.00 | 19.12 | -0.81 | 4.38 |
| GLM-4-9B-Chat | 0.00 | 23.33 | 1.32 | 0.13 |
| Vicuna-v1.5-7B | 0.00 | 14.28 | 2.00 | 0.00 |
| Galactica-1.3B | 0.00 | 33.55 | 0.01 | 0.24 |
| *Closed-Source LLM* | | | | |
| GPT-4o-mini | -0.39 | 30.37 | 1.59 | 1.22 |
| GPT-4o | 0.00 | 0.00 | -3.29 | 1.17 |
| *Specific Biology LLM* | | | | |
| InstructProtein-1.3B | 0.00 | 5.58 | 1.53 | -1.55 |
| Llama-molinst-protein-7B (Mol-Ins) | 0.00 | 13.85 | -1.38 | 3.71 |
| BioMedGPT-LM-7B | 0.00 | 19.71 | 0.92 | -2.39 |
| *Our Model on Balanced Dataset* | | | | |
| ours (stage 1 + balanced stage 2) | 4.13 | 42.92 | -1.48 | 8.29 |
| *Our Model on Our Dataset* | | | | |
| ours (stage 2 only) | 4.77 | 4.25 | 0.72 | 1.61 |
| ours (stage 1 + stage 2) | 1.68 | 56.31 | 10.26 | 70.80 |
| ours (stage 1 + stage 2 + stage 3) | 3.37 | 56.25 | 1.06 | 74.26 |

Table 8: Hints for each task

| Task | Hint |
|------|------|
| Epigenetic Marks Prediction | Return yes or no. |
| Promoter Detection | Return yes or no. |
| Core Promoter Detection | Return yes or no. |
| Enhancer-Promoter Interaction Prediction | Return yes or no. |
| RNA-Protein Interaction Prediction | Return yes or no. |
| Antibody-Antigen Neutralization | Return yes or no. |
| Transcription Binding Sites Detection Human | Return yes or no. |
| Transcription Binding Sites Detection Mouse | Return yes or no. |
| EA Prediction | Return two numeric values with two decimal places for 'Housekeeping EA' and 'Developmental EA'. |
| Fluorescence Prediction | Return one numeric value with two decimal places. |
| Enzyme Commission Number Prediction | Return Enzyme Commission number(s), e.g., 2.7.11.12 |
| Solubility Prediction | Return yes or no. |
| Stability Prediction | Return one numeric value with two decimal places. |
| Thermostability Prediction | Return one numeric value with two decimal places. |
| APA Isoform Prediction | Return one numeric value with two decimal places. |
| Non-coding RNA Function Classification | Return one RNA class: 5S_rRNA, 5_8S_rRNA, tRNA, ribozyme, CD-box, miRNA, Intron_gpI, Intron_gpII, HACA-box, riboswitch, IRES, leader, or scaRNA. |
| Modification | Return RNA modification(s): Am, Cm, Gm, Um, m1A, m5C, m5U, m6A, m6Am, m7G, Psi, AtoI, or none. |
| Mean Ribosome Loading Prediction | Return a numeric value with two decimal places. |
| Programmable RNA Switches | Return three numeric values with two decimal places for 'ON', 'OFF', and 'ON/OFF'. |
| CRISPR On Target Prediction | Return a numeric value with two decimal places. |
| siRNA Efficiency Prediction | Return a numeric value with two decimal places. |

Table 9: Prompt format for baselines

**Prompt format for open-source LLMs:**

My question is {input} This is a {task_type} task. {hint} Do not explain or repeat.

**Prompt format for closed-source LLMs:**

You are an expert biology AI assistant specializing in sequence-related topics. Focus on: DNA, RNA, and protein sequences When answering questions, please follow this format: First give a direct answer in JSON dict such as: {"answer": "Yes"}:

Remember to follow the provided rules:
- For binary classification questions: Answer "Yes" or "No".
- For multi-label classification questions: State the specific label(s).
- For regression questions: Provide the numerical value or range.

Answer the question: "{input}".
Task type: {task_type}.
For better understanding the task, hint: {hint}.

# D    DATA QUALITY CONTROL FOR STAGE 3 REASONING DATA

To ensure the quality and reliability of Stage 3 reasoning data, we have established a robust multi-step validation process:

## D.1    SELF-VALIDATION BY THE MODEL

Once the data is generated, the large language model conducts a self-check to ensure compliance with four core criteria outlined in the data generation prompt, as illustrated in Figure 10:

- Providing a detailed and accurate analysis of the sequence
- Accurately recalling task-related knowledge from studies, databases, or academic sources;
- Engaging in comprehensive reasoning to draw logical conclusions for the question
- Citing relevant references where applicable. The model is required to output the results of its self-check and provide recommendations for improvement in cases that do not meet the standards

For outputs that fail to meet these criteria, specific issues are identified, and the model is instructed to regenerate outputs that adhere to the required standards based on the evaluation results.

## D.2    SECONDARY REVIEW BY AN INDEPENDENT MODEL

Following the initial validation, a second large language model, Gemini-1.5-pro, is employed to independently review and verify the accuracy and consistency of the reasoning paths. Additionally, GPT4o-mini is tasked with reconstructing any unqualified cases based on feedback from Gemini-1.5-pro.

This rigorous quality assurance process not only ensures the integrity of the data but also lays a strong foundation of high-quality training data, enhancing interpretability in downstream tasks.

---

**System Prompt**

<role>You are an expert biology AI assistant specializing in sequence-related topics. Your task is to provide clear, accurate, and comprehensive answers to biology questions, with a focus on:

- DNA, RNA, and protein sequences
- Bioinformatics and computational biology
- Molecular biology techniques
- Evolutionary analysis
- Genomics and proteomics</role>

<task>Carefully read and analyze a given biology sequence-related question and its corresponding standard answer. After thoroughly understanding both, rewrite and enhance the answer.</task>

<requirements>

1. Question Analysis:
   - Identify the question type (e.g., yes/no, multiple choice, open-ended)
   - Determine the specific biological topic and concepts involved
2. Sequence Examination:
   - Analyze the provided biological sequence
   - Identify its omics classification (genomics, transcriptomics, proteomics, etc.)
   - Investigate potential functions and features in nucleotide or amino acid level
3. Contextual Research:
   - Explore relevant biological information connected to the question and sequence
   - Consider all related studies, databases, or academic sources that you can remember
4. Answer Synthesis:
   - Integrate the standard answer with insights from your analysis
   - Organize information in a logical, coherent manner
5. Enhanced Explanation:
   - Prepare a clear, detailed explanation of the biological concepts involved
   - Include a step-by-step chain of thought reasoning process if possible
   - Use precise scientific terminology appropriately
6. Quality Check:
   - Ensure scientific accuracy and depth
   - Verify that the response is clear, engaging, and tailored to the question
   - Confirm that all key points from the original answer are addressed and expanded upon

Provide ONLY the enhanced answer to the biology sequence-related question. Your response should:

- Begin with a rich analysis of the biological sequence, highlighting key functions and features and specific nucleotide or amino acid
- Present the answer in a natural, readable style
- Provide any relevant knowledges and information from studies, databases, or academic sources
- Support your response with a comprehensive explanation
- Outline your thought process clearly even if it might be wrong
- Cite relevant studies, databases, or academic sources when applicable
- Do not include any of the analysis steps or thought processes in your output. The response should be the final, polished answer ready for the end-user to read and understand.
  Especially do not include any information about the standard answer, assume you are providing a answer directly from your own analysises!

</requirements>

---

**Example Prompt**

Please follow the system prompt and given the question:
CCTTCACCGCCACACGGTCCCACTGGACGCATCACTTTTTCGGTAGCAAAAACTTGTAGAAAATTGTCA
ATTCAATTCGCTGGTAGTTGTATATCCATGTGAAGCAGAGATAAGCTCCGCCAACTCTACCCGTCGCCC
ACGCCACGTCGCACAGTTGCGCCCCGCCACAAATATCGAAACCAGGCCTCACAGACCGCAAGCATAC
CCCAAAATGAGCTCCTCGAATGCGGGACAGCGCACGAAGCTAAT Could you run an enhancer activity simulation for this sequence?.

Answer to be refined: {"hk":"4.66","dev":"2.1"}

Your refinement:

Figure 10: An example of a prompt used to generate reasoning data. The system prompt outlines the requirements for the data construction task for GPT-4o-mini. Answers are refined, and corresponding questions are placed within specific prompts.

Table 10: Examples of question and answer template pairs in stage 2 training data.

| Task | Question template | Answer template |
|------|-------------------|-----------------|
| Epigenetic Marks Prediction | `<dna>`{DNA}`</dna>` Are there any characteristic epigenetic marks in this DNA? | After careful EMP analysis, there is conclusive evidence of epigenetic marks in the given DNA sequence. (Positive case) |
| Core Promoter Detection | `<dna>`{DNA}`</dna>`: Evaluate this sequence for potential promoter regions. | No, a promoter region is not present in the given genomic fragment. (Negative case) |
| Enhancer Activity Prediction | `<dna>`{DNA}`</dna>` Enhancer activity in this sequence - what's the deal? | The enhancer activity prediction yields: HK - {hk_enrichment}, Dev - {dev_enrichment} |
| CRISPR On Target Prediction | `<rna>`{RNA}`<rna>` What is the predicted on-target activity of the sequence? | The sequence has an on-target efficiency score of {label}. |
| Programmable RNA Switches | `<rna>`{RNA}`<rna>` What ON/OFF ratio can be expected from the sequence? | The ON state for this sequence is {label_ON}, the OFF state is {label_OFF}, and the ON/OFF ratio is {label_ON_OFF}. |
| Modification Prediction | `<rna>`{RNA}`<rna>` [RNA modification classification] Which RNA modifications are inferred from the sequence? | The RNA modifications for the sequence are predicted as {label}. |
| Fluorescence Prediction | `<protein>`{protein}`<protein>` How effective is the predicted fluorescence for this GFP sequence? | The GFP's fluorescence is calculated to be {label}. |
| Enzyme Commission Number Prediction | `<protein>`{protein}`<protein>` What is the specific activity of this protein as an enzyme? | Specific enzyme activities are defined by EC number {label}. |
| Solubility Prediction | `<protein>`{protein}`<protein>` Can solubility be confirmed for this protein? | This protein shows positive solubility. (Positive case) |
| Enhancer-Promoter Interaction Prediction | `<dna>`{DNA1}`<dna>` promoter + `<dna>`{DNA2}`<dna>` enhancer: connected? | We find no support for promoter-enhancer communication in this genetic material. (Negative case) |
| Antibody-Antigen Neutralization | Is there any molecular recognition between this Ab-Ag pair: `<protein>`{AB}`<protein>>`-`<protein>`{AG}`<protein>>`? | The computational predictions do not support any significant Ab-Ag interaction. (Negative case) |
| siRNA Efficiency Prediction | `<rna>`{SiRNA}`<rna>` `<dna>`{targetDNA}`<dna>` How does the target RNA perform in terms of siRNA-induced mRNA reduction? | The siRNA treatment results in an mRNA remaining percentage of {label}. |

Table 11: An example of question and answer pairs in stage 3 training data for siRNA Efficiency Prediction task.

| Task: siRNA Efficiency Prediction | |
| --- | --- |
| **Question** | `<rna>{SiRNA}<rna>` `<dna>{targetDNA}<dna>` How much mRNA remains after siRNA treatment of the target RNA? |
| **Answer** | To assess the amount of mRNA remaining after siRNA treatment, we first need to understand the mechanisms involved in RNA interference (RNAi) and its impact on mRNA levels. |
| | siRNA (small interfering RNA) is a class of double-stranded RNA that can lead to the degradation of complementary mRNA molecules, effectively silencing their expression. When siRNA binds to target mRNA, it recruits the RNA-induced silencing complex (RISC), which then cleaves the mRNA, reducing its stability and leading to its degradation. |
| | To determine how much mRNA remains after treatment, one typically considers: 
 1. The initial concentration of mRNA. 
 2. The efficacy of siRNA in degrading the target mRNA. 
 3. The kinetic parameters related to both the mRNA decay rate and the siRNA efficacy. |
| | For instance, if initial studies or assays show that treatment with specific siRNA reduces target mRNA levels by approximately 80% under standard experimental conditions, then we can assume that 20% of the original mRNA remains. However, the problem states that the final calculated remaining mRNA is estimated to be 80.6% of the original mRNA. |
| | This indicates that the siRNA treatment was not completely effective, or there might be factors such as incomplete targeting or secondary structures in the mRNA that protected it from degradation. |
| | In summary, after siRNA treatment of the target RNA, the remaining amount of mRNA is therefore 80.6%, indicating a less than complete silencing effect. This percentage reflects the balance of mRNA decay, the effectiveness of the siRNA treatment, and the biological context in which the treatment was performed. |

Table 12: An example of question and answer pairs in stage 3 training data for Noncoding RNA Family Prediction task.

| Task: Noncoding RNA Family Prediction | |
|---|---|
| **Question** | `<rna>{RNA}<rna>` How does this RNA sequence contribute to gene regulation? |
| **Answer** | This RNA sequence represents a component that likely plays a crucial role in gene regulation, particularly through its classification as a "leader" RNA. In the context of molecular biology, leader RNAs are typically found at the 5' end of mRNA sequences and can influence gene expression by engaging in regulatory interactions with ribosomes or other molecular partners.

Leader sequences can have multiple functions, including:
1. **Regulatory Functions:** They may harbor sequences that can bind small regulatory molecules or proteins, thus modulating the transcription or translation of the downstream coding region.
2. **Secondary Structures:** The folding patterns in leader sequences can form intricate secondary structures that contribute to transcriptional attenuation or the formation of a riboswitch. These structures can determine whether the RNA remains in a transcriptionally active state or folds into a configuration that halts transcription or translation.
3. **Interaction with Ribosomes:** The leader region often acts as a signal for ribosomes to initiate translation. For mRNAs, the presence of specific motifs within this region can influence how efficiently translation occurs, impacting overall protein synthesis levels and thus the expression of the associated gene.

In summary, the RNA sequence presented can be classified as a leader, contributing to gene regulation through its interactions and structural characteristics. This underscores the importance of untranslated regions (UTRs) in controlling gene expression. Understanding such RNA sequences and their functions provides valuable insights into the mechanisms of gene regulation, highlighting how sequences beyond coding regions play vital roles in cellular processes. |

1674
1675
1676
1677
1678
1679
1680
1681
1682
1683
1684
1685
1686
1687
1688
1689
1690
1691
1692
1693
1694
1695
1696
1697
1698
1699
1700
1701
1702
1703
1704
1705
1706
1707
1708
1709
1710
1711
1712
1713
1714
1715
1716
1717
1718
1719
1720
1721
1722
1723
1724
1725
1726
1727

Table 13: An example of question and answer pairs in stage 3 training data for Core Promoter Detection.

| Task: Core Promoter Detection | |
|---|---|
| **Question** | `<dna>`{DNA}`<dna>` Identify potential core promoter region in this sequence. |
| **Answer** | The provided DNA sequence shows characteristics that warrant careful examination for core promoter elements. Core promoters are essential in initiating transcription and are typically located upstream of the transcription start site. They often contain specific motifs, such as the TATA box, initiator sequences (Inr), and other regulatory elements that facilitate binding of transcription factors and RNA polymerase II.

In the sequence provided, critical components of core promoters might include:
1. **Initiator sequences (Inr):** These sequences often span the transcription start site and can exhibit variability but typically include consensus elements that help delineate the start of transcription.
2. **TATA box:** A common feature among many eukaryotic promoters, which is usually located about 25-30 bases upstream from the transcription start point.
3. **GC-rich regions:** These regions can also play a role in promoter activity by providing binding sites for transcription factors.

Upon examining the sequence:
- There are no clear TATA boxes or Inr motifs that align with typical core promoter signatures.
- The sequence appears GC-rich, notably towards the middle, but does not show significant promoter elements consistent with those typically required for core promoter identification.

Given these observations, we can conclude that this sequence does not contain recognizable features indicative of a core promoter region. Therefore, the response to whether a potential core promoter region is present in this sequence is negative. |

