# OpenReview forum: "Biology Instructions: A Dataset and Benchmark for Multi-Omics Sequence Understanding Capability of Large Language Models"
_ICLR.cc/2025/Conference — Submitted to ICLR 2025_

### Official Review · Reviewer_FJUK · 2024-10-18

**Soundness:** 3
**Presentation:** 2
**Contribution:** 3
**Rating:** 5
**Confidence:** 4

**Summary:**

The paper present a new extensive dataset called Biology-Instructions, which aims to enhance the ability of LLMs in comprehending multi-omics biological sequences like DNA, RNA, proteins, and multi-molecules. This dataset supports 21 tasks across various omics types and intends to bridge the gap between LLMs and complex biological sequence-related tasks. Additionally, the authors introduce an impressive baseline model named ChatMultiOmics that utilizes a three-stage training pipeline involving continued pre-training on biological sequences, extensive instruction tuning, and reasoning instruction tuning. Experimental results demonstrate significant performance improvements compared to existing general-purpose and biology-specific LLMs.

**Strengths:**

1. The paper introduces a comprehensive multi-omics instruction-following dataset, which significantly extends the application of LLMs to biological domains.
2. The three-stage training pipeline for ChatMultiOmics addresses existing limitations in instruction tuning for biological tasks.
3. The introduction of Biology-Instructions and ChatMultiOmics could drive advancements in computational biology, particularly in tasks like genetic regulation and therapeutic development. The dataset can potentially become a benchmark for future research.

**Weaknesses:**

1. While the dataset covers many tasks, it does not include all multi-omics interactions, and more complex regulatory networks are underrepresented. This limits the scope of the model’s generalizability.
2. The training process is impacted by task imbalances, which could affect the model’s ability to handle underrepresented tasks.
3. The third stage of training (reasoning instruction tuning) showed adverse effects on some regression tasks, indicating that further optimization of the training process is necessary.

**Questions:**

1. Given the challenges posed by task imbalance in the dataset, does the trained model exists bias?
2. Could you provide more insights into why reasoning instruction tuning negatively impacted some regression tasks? Would a different approach to integrating reasoning data be beneficial?
3. What is main differerence compared with other bio-LLM dataset? such as Mol-Instruction. Your dataset construction process is very similar to Mol-Instruction.
4. Can you provide more details of data quality control process? especially for the reasoning data.
5. How about the performance of task-specific models? such as the conventional SOTA model for RNA-Protein Interaction Prediction. How does ChatMultiOmics compare to it?

---

> ### Author Response · Authors · 2024-11-28
> **Answer to Reviewer FJUK**
>
> ### **Weakness1: Multi-omics Interactions**
>
> Thank you for pointing out this concern. In this work, our primary focus is on tasks involving multi-molecular interactions, encompassing both intra-modal and cross-modal tasks. Our benchmark already includes tasks that explore interactions between different molecular modalities, enhancing its diversity:
>
> RNA-Protein Interaction Prediction: Investigating interactions between ncRNAs and proteins can deepen our understanding of the biological mechanisms involving ncRNAs.
> siRNA Efficiency Prediction: Examining the silencing efficiency of siRNA-DNA interactions supports the development of more effective gene therapy drugs.
> In the future, we plan to further expand the range of multimodal tasks, enabling models like ChatMultiOmics to gain a more comprehensive understanding of molecular and cellular biology.
>
> ### **Weakness2: Dataset Balance**
> We conducted experiments on this question. For further information, please refer to `Global response 3.Data imbalance in Biology-Instructions`.
>
> ### **Weakness3: Effect of Stage 3 training**
>
> We appreciate your valuable suggestions regarding the data from the third stage. As you have pointed out, training in the third stage has indeed had a negative impact on some regression tasks. We speculate that this may be due to the following reasons:
>
> Complexity of Large Model Construction: The data in the third stage is generated by the large language model. For the model, interpreting regression tasks may be more challenging than classification tasks. Despite providing relevant information for each task in the hints, the model may still struggle with specific analyses. This often results in the model mentioning generalized information like, "For further validation, one might reference databases such as ENCODE," rather than conducting targeted analysis. We conduct multiple rounds of quality control to filter these data out; however, the data for regression tasks are still not fully desirable.
>
> Inherent Challenges of Regression Tasks: Training a large model to handle regression tasks is inherently challenging. Regression tasks require the model to make accurate numerical predictions, and the additional textual information may negatively impact the training process, exacerbating this difficulty. We will continue to review and optimize the data construction and training process in the third stage to mitigate any adverse effects on regression tasks and provide more effective model support for these tasks.
>
> ### **Weakness4: Compare to Other Bio-LLM Dataset**
> We present the first large-scale multi-omics instruction fine-tuning dataset constructed around the central dogma of molecular biology. In terms of task design, our dataset is unique, encompassing tasks related to DNA, RNA, and proteins, as well as tasks involving multi-molecular interactions. In contrast, Mol-Instruction [r1] is an excellent work focused on molecular biology, with its tasks primarily revolving around molecule-related tasks, protein-related tasks, and molecule-related textual data. This results in a substantial difference in task coverage between our dataset and Mol-Instruction.
>
> Regarding dataset construction, the approach used in Mol-Instruction and similar works is a well-established and proven methodology. Therefore, we have drawn inspiration from their construction methods while designing our task set to suit the multi-omics domain. Methodologically, such adaptation is both reasonable and necessary, as it ensures the reliability of the dataset construction process and provides a strong foundation for our research goals. However, our contribution lies in extending this framework to encompass multi-omics tasks centered on the central dogma, thereby addressing a gap in existing datasets. Additionally, we note that we are the first to integrate reasoning data in Bio-LLM datasets.
>
> We believe that the Biology-Instructions dataset establishes a novel benchmark for research in genomics, transcriptomics, proteomics, and their interactions. This not only complements the scope of Mol-Instruction but also paves the way for the further development of biological large language models.
>
> ### **Weakness5: Data quality control**
>
> We conducted quality control of the dataset in two main stages:
>
> #### Quality Control for Stage 2 Dataset
> At this stage, we focused primarily on the quality control of question-answer templates. As mentioned in the main text, we ensured diversity in the grammar and style of the data templates and constructed a sufficient number of question-answer template pairs. Additionally, to further ensure the quality of the templates, we required task template authors to perform cross-checking to ensure that all templates met the predefined requirements and achieved high-quality standards.
>
> #### Quality Control for Stage 3 Dataset
> For specific information , please refer to `Global response 2.Data Quality Control for Stage 3 Reasoning Data`

---

> ### Author Response · Authors · 2024-11-28
> **Answer to Reviewer FJUK**
>
> ### **Weakness6: Comparison with Task-Specific Models**
>
> **TABLE r1：Comparison with SOTA in DNA Tasks**
> | Model\Task | Enhancer activity (hk) | Enhancer activity (dev) | EMP | TF-H | TF-M | PD | CPD |
> |---|---|---|---|---|---|---|---|
> | Literature SOTA | DeepSTARR| DeepSTARR| DeepSTARR | DNABERT2|DNABERT2 | DNABERT2 | DNABERT2|DNABERT2 |
> | | 68 | 74 | 58.83 | 66.84 | 71.21 | 83.81 | 71.07 |
> | Ours | 55.25 | 44.93 | 5.03 | 22.3 | 32.21 | 56.13 | 44.19 |
>
>
> **TABLE r2: Comparison with SOTA in RNA Tasks**
> | Model/Task         | Isoform | ncRNA  | Modification | MRL    | PRS    | CRISPR On Tar |
> |--------------------|---------|--------|--------------|--------|--------|----------------|
> | Literature SOTA    | APARENT | APARENT | GCN          | GCN    | MultiRM | MultiRM        |
> |                    | 50.82   | 85.73  | 84           | 78     | 55.67  | 44.1           |
> | Ours               | 59.01   | 63.09  | 59.06        | 47.64  | 26.57  | -0.02          |
>
> **TABLE r3: Comparison with SOTA in Protein Tasks**
> | Model/Task         | Function EC     | Stability      | Fluorescence   | Solubility     | Thermostability |
> |--------------------|-----------------|----------------|----------------|----------------|------------------|
> | Literature SOTA    | SaProt-GearNet  | Evoformer      | Shallow CNN    | DeepSol        | ESM-1v          |
> |                    | 88.9           | 79             | 69             | 77             | 78              |
> | Ours               | 19.79          | 60.25          | 2.57           | 63.02          | 45.07           |
>
> **TABLE r4: Comparison with SOTA in Multi-Molecule Tasks**
> | Model/Task         | Enhancer-Promoter | siRNA   | Antibody-Antigen | RNA-Protein     |
> |--------------------|-------------------|---------|------------------|-----------------|
> | Literature SOTA    | EPI-DLMH          |    -    | DeepAAI          | ncRPI-LGAT      |
> |                    | 53.59             |    -    | 54.9             | 93.2            |
> | Ours               | 3.37              | 56.25   | 1.06             | 74.26           |
>
> **TABLE r5: Comparison Between DNABERT2 and Our Model Trained on GUE Benchmark**
> | Model/Task   | EMP | TF-H | TF-M | PD | CPD |
> |-----|------|------|------|------|-------|
> | DNABERT2 | 58.03 | 66.84	| 71.21 | 83.81 | 71.07 |
> | Ours  | 19.16 | 61.62 | 66.11 | 77 | 66.9 |
>
> As shown in TABLE r1-4, ChatMultiOmics currently exhibits a significant performance gap compared to specialized models. We attribute this to the following reasons:
>
> - Specialized models are often fine-tuned with task-specific heads, enabling them to excel at individual tasks. In contrast, training a single model to handle multiple tasks simultaneously introduces significant complexity due to the need for broader generalization. Despite these challenges, our approach successfully integrates 21 tasks into a single unified model.
>
> - Encoder-only models, particularly those based on the BERT architecture, demonstrate exceptional capabilities in biological sequence understanding tasks, resulting in superior performance on specialized tasks. Notably, current SOTA models[r2, r3] predominantly rely on encoder-only architectures, which contribute to their optimal performance compared with decoder-only model like HyenaDNA[r4].
>
> - The amount of pre-training data available for ChatMultiOmics is substantially smaller than that for models like DNABERT2. To achieve comparable training data sizes for large language models, such as the notably larger Llama-3.1-8B, significantly greater computational resources are required. We believe that further extensive pre-training of ChatMultiOmics could effectively enhance its performance. To support this claim, we conducted experiments on the GUE benchmark, comparing the performance of encoder-only models to the Llama3.1-8B model without instruction tuning. The specific settings are: 1. Training was performed using the GUE dataset without any instruction tuning; 2. The model utilized the same pre-trained checkpoint as ChatMultiOmics; 3. Share the same training setting such as learning rate weight ChatMultiOmics. The results are shown in TABLE r5.
>
> - LoRA [r5] shows suboptimal performance in tasks involving biological sequence understanding. While full fine-tuning could potentially yield better results, it requires significantly longer computational time due to increased communication demands compared to LoRA.
>
> Based on the above factors, we acknowledge the considerable performance gap between ChatMultiOmics and specialized models. However, we believe that with better training paradigms and additional computational resources, ChatMultiOmics could achieve results comparable to those of specialized models in the future.
>
> That said, the primary focus of this work is not on achieving SOTA performance but on the construction of the Biology-Instructions dataset. We  hope that future research will leverage our dataset to develop more powerful biological large language models.

---

> ### Author Response · Authors · 2024-11-28
> **Answer to Reviewer FJUK**
>
> [r1] Mol-Instructions: A Large-Scale Biomolecular Instruction Dataset for Large Language Models. ICLR 2024.
>
> [r2] DNABERT-2: Efficient Foundation Model and Benchmark For Multi-Species Genomes. ICLR 2024.
>
> [r3] The Nucleotide Transformer: Building and Evaluating Robust Foundation Models for Human Genomics. bioRxiv 2023.
>
> [r4] HyenaDNA: Long-Range Genomic Sequence Modeling at Single Nucleotide Resolution. NeuralIPS 2023.
>
> [r5] LoRA: Low-Rank Adaptation of Large Language Models. ICLR 2022.

---

### Official Review · Reviewer_AZVG · 2024-10-27

**Soundness:** 2
**Presentation:** 3
**Contribution:** 3
**Rating:** 3
**Confidence:** 5

**Summary:**

This paper introduces a new dataset and model, Biology Instructions and ChatMultiOmics, designed for multi-modal biological data. The dataset distinguishes itself by unifying DNA, RNA, and protein data types, which were previously treated separately. The model undergoes three stages of training: continual pretraining, supervised fine-tuning (SFT), and reasoning training. After training on the Biology Instructions dataset, ChatMultiOmics demonstrates superior performance over state-of-the-art baselines on 21 tasks curated by the authors.

**Strengths:**

1-Unified Multi-Omics Dataset: By integrating DNA, RNA, and protein sequences within a single dataset, this work allows ChatMultiOmics to leverage a more comprehensive representation of biological data. This approach enables the model to capture cross-modal relationships that traditional, single-modal datasets cannot, potentially enhancing predictive power in complex biological systems.

2-Performance on Custom Benchmark: The model demonstrates improved performance on 21 tasks specifically designed by the authors. These tasks, which span classification, regression, and possibly other benchmarks within the domain, provide a new set of evaluation standards. The demonstrated gains in performance could serve as a reference point for future biological language models seeking to handle multiple data types concurrently.

3-Empirical Training Insights: The work contributes an empirical analysis of training methodologies suited to biological data. By introducing three distinct stages of training—continual pretraining, supervised fine-tuning, and reasoning training—the authors outline a potentially generalizable approach that may benefit future model development for multi-modal biological applications.

**Weaknesses:**

1-Focus Solely on Predictive Tasks: While the model excels in predictive tasks, the lack of generative experiments, such as protein and DNA design, limits its versatility. Generative tasks are becoming increasingly relevant, where designing novel sequences with specific functionalities is critical. Including generative experiments could enhance the paper by showing the model's potential in creating or optimizing biological sequences. The generative task is of the highest interest.

2-Lack of Structural Data Integration: The model currently relies only on sequence data as input, which, while valuable, is limited. In biological systems, structural information is crucial, as the 3D structure of molecules like proteins often determines their function. Incorporating structure data, such as 3D coordinates from protein databases (e.g., PDB), could enhance predictive capabilities.

3-Evaluation Limited to In-House Dataset: The performance metrics presented are limited to the authors' own test dataset, which limits the generalizability of the results. Reporting the model's performance on established, domain-specific datasets in DNA and protein domains, such as those used in ChatNT, ProLLaMA, and InstructProtein, would add robustness to the claims of improvement and allow a better assessment of the model's comparative advantages. Otherwise, it is unfair to judge the proposed model with others.

4-Minor language issues: in Line 315, "We" should be "we"; in Line 290, "Figure 10" should be "Table 10."

**Questions:**

1-Definition of Reasoning in Stage 3: Could you clarify what "reasoning" entails in the context of Stage 3 training? Specifically, how does reasoning differ from the classification and regression tasks in Stage 2? "Step-by-step" prompting alone does not fully justify the use of the term "reasoning," as true reasoning often involves complex inferencing, such as multi-hop or chain-of-thought reasoning. Did you explore techniques like chain-of-thought prompting, where the model reasons through intermediate steps, or alternative strategies that might better align with reasoning tasks?

2-IChoice of LoRA+ vs. Full-Parameter Pretraining in Stage 1: I am curious about the decision to use LoRA+ for continual pretraining instead of a full-parameter approach. Full-parameter pretraining could potentially offer stronger alignment with downstream tasks by updating the entire model. Additionally, I wonder if incorporating text data (e.g., biological literature or annotations) in Stage 1 could mitigate forgetting and enhance the model's knowledge base for better contextual understanding. This could enrich the model with domain knowledge without the constraints of task-specific fine-tuning alone.

3-Limiting Data to Human-Specific DNA and RNA: What was the rationale for restricting DNA and RNA data to human species only? Incorporating non-human data, including model organisms (e.g., E. coli, C. elegans, D. melanogaster), could improve the model's generalization and relevance to broader biological contexts, especially in fields like comparative genomics and evolutionary studies.

4-Enhancing Dataset Diversity with Multi-Hop Connections: The current dataset lacks multi-hop connections, which could limit the model’s ability to perform complex reasoning over interconnected data points. Mixing this dataset with established datasets from different domains, or incorporating multi-hop relational data, could enhance diversity and enable the model to capture deeper biological relationships. I am particularly interested in the potential for training another model with such a mixed dataset, as this could yield a more robust model capable of handling complex biological queries across domains.

---

> ### Author Response · Authors · 2024-11-28
> **Answer to Reviewer AZVG**
>
> ### **Weakness 1: Focus Solely on Predictive Tasks**
>
> We appreciate this valuable feedback and would like to take the opportunity to clarify the intent and scope of our work. Our study intends to focus on evaluating tasks on **sequence understanding** through, specifically assessing how instruction-tuned language models, designed for general-purpose tasks, can perform on biological sequence-based prediction tasks.
>
> While generative tasks, such as sequence design, are indeed important and promising, they are beyond the scope of this study. Our primary goal is to integrate multiple omics data and provide a comprehensive benchmark from a biological perspective, rather than aiming for versatility across both predictive and generative task types. Incorporating generative experiments would require distinct evaluation methodologies, which are not addressed in this work.
>
> We recognize the value of generative tasks and view them as an important direction for future research. We plan to explore generative applications in subsequent work based on insights gained from this benchmarking effort. To ensure clarity, we have revised the paper to more explicitly define the scope of this study in the `Abstract`, `Section 1 Introduction`, `Section 3.2.1 Tasks` and included generative tasks as a future direction in `Section 6 Discussion`, highlighted in `blue`.
>
> ### **Weakness 2: Lack of Structural Data Integration**
>
> We appreciate this advice and agree on the importance of molecular structural data in functional prediction tasks. However, incorporating structural data to develop a method that excels in function prediction is not our primary intention for this work.
>
> The core research question we aim to answer is whether instruction-tuned language models, which are proficient in understanding human language, can also demonstrate strong performance in biological **sequence** understanding to address biologically critical tasks. The motivation guiding us to select our tasks and dataset stems from the intrinsic similarity between biological sequence data and human language—both being discrete and sequential in nature. Incorporating structural data, which is continuous in nature, represents a distinct research direction that falls beyond the scope of this work. Our contribution lies in establishing a benchmark and dataset that covers multiple omics in biology, serving as a foundation for the research community to advance multi-omics studies.
>
> We acknowledge the value of structural data and its potential to complement sequence-based approaches. As such, we have included the integration of structural data as a future direction in `Section 6 Discussion` to explore in subsequent studies, highlighted in `blue`.

---

> > ### Author Response · Authors · 2024-11-28
> > **Answer to Reviewer AZVG**
> >
> > ### **Weakness 3: Evaluation Limited to In-House Dataset**
> >
> > We appreciate this insightful feedback and agree that evaluating the model on established, domain-specific datasets would provide a more robust and fair assessment of its performance. Notably, some of our datasets, such as GUE, are adapted from datasets used in the mentioned models, ensuring that our comparisons remain fair and meaningful. Our work primarily focuses on the contribution of benchmarks and datasets, with our proposed method serving as a baseline to underscore the promising potential of multi-omics research.
> >
> > We acknowledge that relying solely on our in-house dataset may create an impression of bias. While we have considered testing on datasets from other models, several challenges have limited our ability to conduct these experiments. For instance, the dataset used in ChatNT is not publicly available, and models like ProLLaMA primarily target protein design and generative tasks, which diverge significantly from our focus on multi-omics sequence understanding tasks. Furthermore, the time and resources required to adapt and integrate datasets from such diverse domains into our framework present practical challenges.
> >
> > Despite these challenges, we are actively working on experimenting with additional datasets and expect to finalize and submit the results in the coming days. We hope it will better address this concern, and strengthen the robustness of our claims and further validate the generalizability of our findings.

---

> > > ### Author Response · Authors · 2024-11-28
> > > **Answer to Reviewer AZVG**
> > >
> > > ### **Weakness 4: Minor Languages Issues**
> > >
> > > Thank you for pointing out these issues. We have corrected the typo in line 315 and highlighted in `red`. Regarding line 290, we confirm that the reference to "Figure 10" is the correct figure we want to refer to. However, we understand the confusion, as Figure 10 is positioned just above Table 10, and the hyperlink might make it appear as if it targets Table 10.
> > >
> > > We greatly appreciate your careful reading and have thoroughly reviewed the manuscript to ensure clarity and that no other typos remain.
> > >
> > > ### **Question 1: Definition of Reasoning in Stage 3**
> > > In Stage 3, "reasoning" specifically refers to the inclusion of a reasoning component in the training data and the ability of our model to produce answers accompanied by logical, scientific reasoning (examples in `Figure 1 (b)` and `Figure 9`). In Stage 2, training data consists only of labeled instruction templates without reasoning-based outputs. The regression and classification tasks remain consistent across both stages, while the difference is Stage 3 introduces additional reasoning data for training to enhance the model's performance with reasoning capability. Hence, “reasoning” in stage 3 does not refer to “step-by-step” prompting technique in our context.
> > >
> > > During the construction of reasoning data for Stage 3 (detailed in `Section 3.3.2` and `Appendix D Figure 10`), we utilized GPT-4o-mini to generate intermediate reasoning steps through a guided step-by-step approach. This process involved prompting the model to logically infer relationships in supervised data, resulting in curated datasets enriched with chain-of-thought reasoning. These curated datasets enable our model to produce responses that include intermediate reasoning steps.
> > >
> > > ### **An example of stage 2 data(from `Table 10`):**
> > >
> > > **Question**: <rna>{siRNA seq}<rna> <dna>{targetDNA seq}<dna> How does the target RNA perform in terms of siRNA-induced mRNA reduction?
> > >
> > > **Answer**: The siRNA treatment results in an mRNA remaining percentage of 80.6.
> > >
> > > ### **An example of stage 3 data (updated in Appendix `Table11`):**
> > >
> > > **Question**: <rna>{siRNA seq}<rna> <dna>{targetDNA seq}<dna> How much mRNA remains after siRNA treatment of the target RNA?
> > >
> > > **Answer**:
> > > To assess the amount of mRNA remaining after siRNA treatment, we first need to understand the mechanisms involved in RNA interference (RNAi) and its impact on mRNA levels.
> > >
> > > siRNA (small interfering RNA) is a class of double-stranded RNA that can lead to the degradation of complementary mRNA molecules, effectively silencing their expression. When siRNA binds to target mRNA, it recruits the RNA-induced silencing complex (RISC), which then cleaves the mRNA, reducing its stability and leading to its degradation.
> > >
> > > To determine how much mRNA remains after treatment, one typically considers:
> > > 1. The initial concentration of mRNA.
> > > 2. The efficacy of siRNA in degrading the target mRNA.
> > > 3. The kinetic parameters related to both the mRNA decay rate and the siRNA efficacy.
> > >
> > > For instance, if initial studies or assays show that treatment with specific siRNA reduces target mRNA levels by approximately 80% under standard experimental conditions, then we can assume that 20% of the original mRNA remains. However, the problem states that the final calculated remaining mRNA is estimated to be 80.6% of the original mRNA.
> > >
> > > This indicates that the siRNA treatment was not completely effective, or there might be factors such as incomplete targeting or secondary structures in the mRNA that protected it from degradation.
> > >
> > > In summary, after siRNA treatment of the target RNA, the remaining amount of mRNA is therefore 80.6%, indicating a less than complete silencing effect. This percentage reflects the balance of mRNA decay, the effectiveness of the siRNA treatment, and the biological context in which the treatment was performed.

---

> > > > ### Author Response · Authors · 2024-11-28
> > > > **Answer to Reviewer AZVG**
> > > >
> > > > ### **Question 2: Choice of LoRA+ vs. Full-Parameter Pretraining in Stage 1**
> > > >
> > > > We fully recognize that the LoRA+ method may result in suboptimal performance compared to full fine-tuning, especially for tasks like biological sequence understanding, where large language models lack sufficient pretraining. However, the primary reason we opted for LoRA+ instead of full fine-tuning lies in the limitations of computational resources. Even with multi-GPU distributed training minimizing the memory requirements for full fine-tuning, it still demands significantly longer training time, which we attribute mainly to the additional optimizer state communication overhead. Under these resource constraints, we chose to adopt different fine-tuning strategies at different stages to minimize resource consumption. Nevertheless, we would like to emphasize that the core focus of this work is the Biology-Instructions dataset itself.
> > > >
> > > > We fully agree that incorporating relevant text data during the pretraining phase or adding text-based instruction fine-tuning data to the Biology-Instructions dataset can improve the model’s understanding of related tasks and lead to better results. However, we have not pursued this direction in this work primarily because we are uncertain whether introducing text data unrelated to the central dogma would positively impact model performance. At the same time, the amount of text data directly related to the central dogma is currently insufficient to make a significant impact. To better reponse your instrest, we are conducting experiments for integrate PubMedQA dataset in our instruction-tuning dataset to expore further optimization, and we will reply the result very soon! Thank you again for your valuable comment on the training strategy and data set composition.

---

> > > > > ### Author Response · Authors · 2024-11-28
> > > > > **Answer to Reviewer AZVG**
> > > > >
> > > > > ### **Question 3: Limiting Data to Human-Specific DNA and RNA**
> > > > > - **Diverse Species in DNA and RNA Tasks**: Contrary to the assumption that our DNA and RNA tasks are exclusively based on human data, we also include datasets from diverse species. For instance, the Epigenetic Marks Prediction dataset is based on `yeast`, the Enhancer Activity Prediction dataset uses `fruit fly` data, and the Mouse Transcription Factor Binding dataset is derived from `mouse` data. These datasets contribute to the species diversity within our benchmark’s DNA and RNA tasks.
> > > > >
> > > > > - **Significance of Human Data in Genomic Research**: Studying human gene expression and regulatory mechanisms holds unparalleled biological and medical significance, as humans are the most directly relevant subject in life sciences. Compared to other species, the complexity and specificity of the human genome underline its critical importance in areas such as disease research, precision medicine, and cutting-edge technologies like gene editing. By focusing on the human genome and its epigenetic regulatory networks, we can uncover molecular mechanisms underlying human health and disease with greater precision, laying the groundwork for transformative medical breakthroughs. Prioritizing human data ensures high-quality and impactful research outcomes, advancing scientific progress to benefit humanity.
> > > > >
> > > > > Looking ahead, we plan to further expand our benchmark to include more species and tasks, enhancing its diversity and enabling deeper insights into evolutionary relationships and other cross-species analyses.
> > > > >
> > > > > ### **Question 4: Enhancing Dataset Diversity with Multi-Hop Connections**
> > > > > Thank you for this insightful feedback. While multi-hop relational data indeed offers significant potential for enhancing model reasoning over complex biological tasks, creating such datasets requires substantial effort and domain expertise, making it less favorable for this study.
> > > > >
> > > > > The goal of developing models capable of capturing deeper biological relationships drives our integration of diverse multi-omics data, the construction of the reasoning dataset, and the design of the three-stage training pipeline, which are our core contributions in this study. This approach enables the model to capture interconnected relationships within the data more efficiently and at scale, as the first step to address this goal.
> > > > >
> > > > > We have acknowledged the promising potential of multi-hop data in the discussion section and included it as a direction for future workn in the manuscript `Section 6 Discussion`, highlighted in `blue`. We are open to collaborating on this avenue to further explore its potential.

---

> > > > > > ### Comment · Reviewer_AZVG · 2024-11-29
> > > > > > **Thank the authors for the rebuttal**
> > > > > >
> > > > > > I have read the rebuttal and thank all the authors for their efforts during the rebuttal stage. Their response does not fully convince me. The scope of this paper is sort of limited and the experimental results are not convincing. The research field of large language models for natural and protein languages needs more thorough, dedicated thinking with solid experiments.

---

> ### Author Response · Authors · 2024-12-02
> **Integrate Text Data to Biology-Insturctions**
>
> Thanks for your feedback. We selected 200k data points from the Biology-Instructions dataset to balance its volume with PubMedQA and conducted experiments using models pre-trained in the first phase. The models were fine-tuned separately on Biology-Instruction-200k and on a combination of Biology-Instruction-200k and PubMedQA. The results indicate that while the inclusion of PubMedQA had a generally negative impact on performance, it showed notable improvements in specific tasks, such as rna_protein_interaction (MCC: 4.44 → 7.17) and thermostability prediction (spearman: 13.69 → 23.27).
>
> | Model                       | Biology-Instruction-200k | Biology-Instruction-200k+PubMedQA |
> |-----------------------------|--------------------------|-----------------------------------|
> | pd (MCC)                    | 27.02                    | 26.61                             |
> | cpd (MCC)                   | 20.35                    | 13.36                             |
> | emp (MCC)                   | 1.28                     | 1.76                              |
> | enhancer_activity (hk_PCC)  | 10.78                    | 6.3                               |
> | enhancer_activity (dev_PCC) | 12.08                    | 8.67                              |
> | tf_h (MCC)                  | 2.61                     | 2.43                              |
> | tf_m (MCC)                  | 7.85                     | 3.06                              |
>
> | Model                         | Biology-Instruction-200k | Biology-Instruction-200k+PubMedQA |
> |-------------------------------|--------------------------|-----------------------------------|
> | NoncodingRNAFamily (Acc)      | 13.78                    | 11.84                             |
> | CRISPROnTarget (spearman)     | 2.94                     | 5.92                              |
> | Isoform (R2)                  | 35.61                    | 28.81                             |
> | Modification (AUC)            | 54.23                    | 53.77                             |
> | MeanRibosomeLoading (R2)      | 0.11                     | 0.03                              |
> | ProgrammableRNASwitches (R2)  | 6.22                     | 4.93                              |
>
> | Model               | Biology-Instruction-200k | Biology-Instruction-200k+PubMedQA |
> |---------------------|--------------------------|-----------------------------------|
> | Stability (spearman)| 0.89                     | 1.44                              |
> | FunctionEC (Fmax)   | 6.39                     | 5.23                              |
> | Solubility (Acc)    | 49.78                    | 51.17                             |
> | Fluorescence (spearman) | -0.3                   | 0.06                              |
> | Thermostability (spearman) | 13.69               | 23.27                             |
>
> | Model                               | Biology-Instruction-200k | Biology-Instruction-200k+PubMedQA |
> |-------------------------------------|--------------------------|-----------------------------------|
> | rna_protein_interaction (MCC)       | 4.44                     | 7.17                              |
> | promoter_enhancer_interaction (MCC) | 4.13                     | 3.37                              |
> | sirnaEfficiency (mixed_score)       | 45.35                    | 43.29                             |
> | antibody_antigen (MCC)              | 0.98                     | -0.37                             |
>
> **1. Positive Impact:**
>
> The inclusion of PubMedQA boosted performance in certain tasks, such as RNA-protein interaction and thermostability prediction. This suggests that PubMedQA may provide complementary knowledge or context for tasks that benefit from enriched textual information, potentially related to functional annotations or interaction studies.
>
> **2. Negative Impact:**
>
> For several other tasks, such as enhancer activity prediction and Isoform prediction, performance decreased when PubMedQA data was included. This might be due to task-specific conflicts or noise introduced by the PubMedQA dataset, which was not curated specifically for these genomic tasks. The textual nature of PubMedQA might have diluted the task-specific signal learned from Biology-Instruction data.
>
> **3. Possible Reasons for Mixed Results:**
>
> - **Task Relevance:** PubMedQA primarily focuses on textual biomedical Q&A, which may not align closely with structured biological tasks like regression or classification on sequence-based features.
> - **Data Volume Balance:** The relatively small size of Biology-Instruction-200k compared to the scale of pre-training data may have made the model more susceptible to overfitting or being biased by PubMedQA's domain.
> - **Domain Compatibility:** Some tasks likely require more structured biological data (e.g., sequence-to-function predictions), whereas PubMedQA introduces noisy textual knowledge that could overshadow task-specific patterns.

---

> > ### Author Response · Authors · 2024-12-03
> > **Integrate Text Data to Biology-Insturctions**
> >
> > While it enriches the model's understanding in interaction-related tasks, careful curation and alignment with task requirements are necessary to avoid performance degradation in more structured tasks. Future work could explore methods to better integrate textual datasets like PubMedQA without compromising on task-specific performance.

---

### Official Review · Reviewer_w8XB · 2024-10-30

**Soundness:** 3
**Presentation:** 4
**Contribution:** 2
**Rating:** 5
**Confidence:** 4

**Summary:**

This paper introduces a large-scale, multi-omics biological sequences-related instruction-tuning dataset and a biologically focused large language model (LLM) using a three-stage training pipeline to incrementally incorporate domain knowledge. The proposed model demonstrates superior performance across 21 biological sequence-related multi-omics tasks compared to state-of-the-art (SOTA) LLMs. The paper is well-structured, easy to follow, and provides numerous technical details.

**Strengths:**

Comprehensive Dataset: The introduction of a large-scale, multi-omics dataset is a valuable resource for the community.
Three-Stage Training Pipeline: The gradual incorporation of domain knowledge through a three-stage training pipeline is a thoughtful approach.
Performance: The model shows superior performance in a variety of tasks, highlighting its potential utility.

**Weaknesses:**

Technical Novelty: While the paper is resource-rich, it offers limited technical novelty. It primarily serves as a resource article.
Training Sample Size: The model is trained on 3 million samples, which is relatively small given the vast space of DNA, RNA, and proteins (in the billions) and the number of parameters in the LLM (8B Llama3.1).
Alignment of Human and Biological Language: The model's alignment of human language with biological language is an interesting concept but could be explored further.

**Questions:**

Figure 7: There appears to be a discrepancy in the results. How can the model without stage 1 and stage 3 (third row) perform much better than the model without stage 1 (second row)? Is this a typo?
Figure 6: The figure is difficult to interpret as the methods are clustered together. A table might provide a clearer comparison.
Insights: The four findings presented seem intuitive and do not offer particularly novel insights.
Mixed Score Formula: The paper should include the formula for the mixed score used in single-label regression to enhance clarity.

---

> ### Comment · Reviewer_w8XB · 2024-11-26
>
> I will keep my score since no response.

---

> ### Author Response · Authors · 2024-11-28
> **Answer to Reviewer w8XB**
>
> Thank you for your feedback. Our primary focus is on addressing the critical need for instruction tuning datasets specifically designed for biological sequence understanding tasks, which are essential for effectively integrating LLMs into multi-omics research. We sincerely thank you for your insightful feedback. Below are our detailed responses to your concerns:
>
> ### **Weakness1: Training Sample Size**
> The 3 million samples refer specifically to the instruction-tuning dataset, which is constrained by the availability of high-quality source data and is therefore both precious and rare. Prior to instruction tuning, we utilized a significantly larger dataset for pre-training to ensure the model learns essential biological knowledge.
>
> Notably, many studies on LLMs in domains such as molecular biology [r1,r2] and mathematics [r3,r4], and general LLM [r5,r6,r7] have demonstrated that instruction tuning on datasets of this scale is highly effective. For example: Mol-Instructions has 2,043,587 instruction-tuning samples across biomolecular tasks, InstructGPT has only 112,801 instruction-tuning samples focused on text-based tasks, and LLaVA[r8] has 158,000 instruction-tuning samples. Compared to these works, our Biology-Instructions dataset, with over 3 million samples, represents one of the largest instruction-tuning datasets in this domain.
>
> Our experiments further confirm that the scale and diversity of Biology-Instructions enhance the model’s performance in biological sequence understanding tasks, aligning with findings from the above studies in other domains.
>
> [r1] Mol-Instructions: A Large-Scale Biomolecular Instruction Dataset for Large Language Models. ICLR 2024.
>
> [r2] InstructMol: Multi-Modal Integration for Building a Versatile and Reliable Molecular Assistant in Drug Discovery. arXiv, 2023.
>
> [r3] MAmmoTH: Building Math Generalist Models through Hybrid Instruction Tuning. arXiv, 2023.
>
> [r4] MathScale: Scaling Instruction Tuning for Mathematical Reasoning. arXiv, 2024.
>
> [r5] Training language models to follow instructions with human feedback. NeurIPS 2022. (InstructGPT)
>
> [r6] Scaling Instruction-Tuned Language Models. JMLR 2024. (Flan-T5)
>
> [r7] Crosslingual Generalization through Multitask Finetuning on a Few Languages. ACL 2023. (BLOOMZ)
>
> [r8] Visual Instruction Tuning. NeurIPS 2023. (LLaVA)
>
> ### **Weakness2: Alignment of Human and Biological Language**
>
> Thank you for highlighting the importance of aligning human and biological language. We agree that this is both a critical and fascinating direction for research. In fact, our study addresses this challenge, as aligning these two domains is essential for using large language models (LLMs) in multi-omics research.
>
> Our experimental results indicate that without pre-training, LLMs struggle to effectively align human and biological language. This gap motivated us to design a three-stage training pipeline to systematically address the alignment challenge. Through this pipeline, our model demonstrates substantial improvements in aligning human and biological language.

---

> ### Author Response · Authors · 2024-11-28
> **Answer to Reviewer w8XB**
>
> ### **Question1: Typo in Figure 7**
> Thank you for identifying this. We confirm that this was indeed a typo in the figure and we have updated it in the manuscript `Figure 7`.
>
> ### **Question2: Detailed Results of Figure 6**
> We appreciate your suggestion to use a table for better clarity. Due to space constraints, we were unable to include detailed tables in the main text. However, we have provided comprehensive tables (`Table 4,5,6 and 7`) in the original manuscript `Appendix`, detailing the results of all methods across all tasks. In the revised version, we have included clearer references to these tables in the main text to guide readers in `green`.
>
> ### **Question3: Insights**
>
> Thank you for your feedback. While we agree that some of the findings may appear intuitive, we believe they are crucial in addressing the unique challenges of applying large language models (LLMs) to biological sequence tasks.
>
> Biological language modeling presents specific challenges due to the unique nature of biological sequences, which differ significantly from natural language. For instance, biological sequences often require specialized tokens and representations to capture their structural and functional information. Although this may seem intuitive, our work is the first to systematically validate this observation through extensive experiments.
>
> We tested a wide range of existing LLMs and consistently found that their performance on biological sequence tasks was inadequate without specialized pre-training and instruction tuning. This suggests that the issue is not isolated to individual models but is a broader limitation across current approaches. Moreover, through these findings, we identified key problems and proposed potential solutions, such as tailored pre-training strategies, instruction datasets, and reasoning-based enhancements, to bridge this gap. These findings provide a foundational understanding for future work.
>
> ### **Question4: Mixed Score Formula**
> Thanks for the suggestion. We have added detailed calculation in `Appendix A.2.1` in `green` for clarity.
>
> The **Mixed Score** is a custom metric adopted from data source [r1] designed to balance regression error and classification accuracy, integrating three components: **Mean Absolute Error (MAE)**, **Range-based MAE (Range-MAE)**, and **F1 Score**. This metric offers a comprehensive evaluation of model performance by considering both overall accuracy and precision within a critical value range.
>
> #### Components
>
> 1. **Mean Absolute Error (MAE)**: Measures the average magnitude of prediction errors across all samples, defined as:
>    $$
>    \text{MAE} = \frac{1}{n} \sum_{i=1}^{n} |y_i - \hat{y}_i|,
>    $$
>    where $n$ is the total number of samples, $y_i$ is the ground truth, and $\hat{y}_i$ is the predicted value. The range of MAE is $[0, 100]$.
>
> 2. **Range-based MAE (Range-MAE)**: Evaluates the MAE within a specific range of interest, focusing on regions where high accuracy is critical. For the siRNA task, the critical range is defined as $[0, 30]$. The formula is:
>    $$
>    \text{Range-MAE} = \frac{1}{m} \sum_{j=1}^{m} |y_j - \hat{y}_j|,
>    $$
>    where $m$ is the number of samples within the range, and $y_j, \hat{y}_j$ are the ground truth and predicted values within this range. Range-MAE is bounded within $[0, 100]$.
>
> 3. **F1 Score**: Combines precision and recall into a harmonic mean for evaluating predictions within the designated range:
>    $$
>    \text{F1} = 2 \cdot \frac{\text{Precision} \cdot \text{Recall}}{\text{Precision} + \text{Recall}}.
>    $$
>
> #### Mixed Score Formula
>
> The final Mixed Score integrates the above components to balance global and range-specific performance:
> $$
> \text{Mixed Score} = 0.5 \cdot (1 - \text{MAE}/100) + 0.5 \cdot \text{F1} \cdot (1 - \text{Range-MAE}/100),
> $$
> where the first term emphasizes overall regression performance (via MAE), and the second term focuses on classification precision and recall within the specified range (via F1 and Range-MAE).
>
> This metric ensures a well-rounded evaluation of model capabilities, rewarding models that perform consistently across global and critical sub-ranges.
>
> [r1] SAIS. sirna data. http://competition.sais.com.cn/competitionDetail/
> 532230/format, 2020. Accessed: 2024-05-26.

---

> ### Author Response · Authors · 2024-11-28
> **Answer to Reviewer w8XB**
>
> Thank you for your time and effort in reviewing our work and for providing valuable feedback. We have been actively working to address your comments, conducting additional experiments, and refining our responses to ensure they are comprehensive and satisfactory. Thank you again for your constructive review and understanding.

---

### Official Review · Reviewer_mgcL · 2024-10-31

**Soundness:** 2
**Presentation:** 3
**Contribution:** 2
**Rating:** 5
**Confidence:** 4

**Summary:**

This paper introduces “Biology-Instructions,” a large-scale multi-omics biological sequence-related instruction-tuning dataset encompassing DNA, RNA, proteins, and multi-molecules. This dataset aims to enhance LLMs’ ability to handle various biology-related tasks. It proposes a new training pipeline, “ChatMultiOmics,” demonstrating improvements in biological sequence understanding.

**Strengths:**

It bridges the gap between LLMs and biological sequence understanding through a new dataset and benchmarking tool called “Biology-Instructions.”

**Weaknesses:**

The study shows that LLMs require specialized pre-training to perform effectively on biology-related tasks, which may not be feasible or practical in all research settings.

Although the paper introduces a novel dataset and training method, it mainly reaffirms the importance of specific pre-training rather than providing new insights into biological sequence understanding.

The dataset may exhibit imbalances where certain categories are more represented than others. This could lead to the model overfitting to tasks with more data during training while neglecting tasks that have less data.

It appears that there is a lack of a data availability statement.

**Questions:**

See weakness.

---

> ### Author Response · Authors · 2024-11-28
> **Answer to Reviewer mgcl**
>
> Dear reviewer, we sincerely thank you for your insightful feedback. Below are our detailed responses to your concerns:
>
> ### **Weakness1: Specialized Pre-training Feasibility**
> We emphasize the unique characteristics of biological sequence tasks. Our experiments on baseline large language models demonstrate that existing open and closed-source models lack pre-training on biological sequences, which is distinctly different from tasks like AI in chemistry [r1]. Although ChatNT [r2] presents a training approach that does not require pre-training, it effectively connects a large language model with an encoder that has already undergone extensive pre-training. This integration is achieved through a percevier-resampler module. However, since there is no well-trained multi-omics encoder model available, we opted not to adopt this method.
>
> Therefore, we highlight the importance of both pre-training and instruction fine-tuning, as they complement each other in achieving optimal training outcomes. We point out that without robust pre-training, even training a model on a large-scale biological sequence instruction fine-tuning dataset would fail to significantly enhance its performance in biological sequence understanding tasks. Conversely, without a large-scale instruction fine-tuning dataset, pre-training loses its value. Given the availability of sufficient resources for biological sequence pre-training data, we propose the Biology-Instructions dataset to bridge the gap in the lack of instruction fine-tuning datasets for biological sequence tasks. Additionally, we validate that incorporating reasoning processes into biological sequence understanding is beneficial for performance and can provide a better interactive experience.
>
> [r1] ChemLLM: A Chemical Large Language Model. arXiv, 2024.
>
> [r2] ChatNT: A Multimodal Conversational Agent for DNA, RNA and Protein Tasks. bioRxiv, 2024.
>
> ### **Weakness2: Dataset Imbalance**
> We appreciate your observation regarding dataset imbalance. We conduct experiment on this question, for further information please refer to `Global response 3.Data imbalance in Biology-Instructions`.
>
> ### **Weakness3: Data Availability Statement**
> Thank you for pointing this out. We confirm that the Biology-Instructions dataset is publicly available. It can now be accessed through an anonymous data link https://anonymous.4open.science/r/Biology-Instructions-FD66/ , also in the revised version of the paper `3.1 Overview of Biology-Instructions` in `green`.

---

### Official Review · Reviewer_Knz7 · 2024-11-04

**Soundness:** 2
**Presentation:** 2
**Contribution:** 2
**Rating:** 5
**Confidence:** 4

**Summary:**

This paper presents "Biology-Instructions," a comprehensive dataset for training LLMs on multi-omics biological sequences. The authors developed ChatMultiOmics, a biology-specific model using a three-stage pipeline: unsupervised pre-training, instruction tuning, and reasoning-based fine-tuning. This approach enhances LLMs' performance in multi-omics tasks while maintaining conversational abilities. Extensive benchmarking shows ChatMultiOmics outperforms general-purpose and some specialized LLMs in sequence-based tasks.

**Strengths:**

Novel Dataset: The introduction of "Biology-Instructions" provides a new, large-scale dataset specifically tailored for multi-omics data (DNA, RNA, proteins, and multi-molecular sequences). This fills a crucial gap, as most current datasets in bioinformatics lack instruction-tuning data suited for large language models (LLMs) in multi-omics contexts.

**Weaknesses:**

1. Lack of Evaluation on Practical Applications: The model is mainly evaluated on isolated multi-omics tasks without a clear link to practical bioinformatics applications (e.g., case study on real-world drug discovery, wet-lab experiments design). This might limit its perceived impact outside academia.

2. Dataset Limitations: The "Biology-Instructions" dataset, while comprehensive for single-modality tasks (DNA, RNA, protein), lacks representation of cross-modality interactions essential for understanding complex biological networks. Limited inclusion of DNA–RNA–protein interactions hinders the model's ability to learn integrated biological insights. Examples of missing interactions include DNA–RNA regulatory loops, DNA–protein binding in gene expression control, RNA–protein associations for post-transcriptional modifications, and multi-molecule complexes like the spliceosome or transcription initiation complex Expanding the dataset to include diverse cross-modality interactions would allow models like ChatMultiOmics to more accurately represent complex cellular processes and enhance its real-world applications.

3. Potential for Data Leakage: The study fails to address or discuss potential data leakage or overlap between training and test sets. This is particularly concerning given the multi-omics nature of the dataset, where DNA, RNA, and proteins are interconnected through the central dogma of molecular biology. Such overlap could significantly impact the validity of the reported performance metrics.

4. Limited Error Analysis: The study lacks an in-depth error analysis, especially for tasks where the model performs poorly. Without this, it's difficult to identify specific weaknesses or areas for improvement in ChatMultiOmics.

5. Cell type-specificity: The "Biology-Instructions" dataset lacks cell-type specificity, a crucial factor for accurate biological modeling. Different cell types have unique gene expression profiles, regulatory networks, and molecular interactions that reflect their specialized functions. Without cell-type-specific data—such as neuron-specific RNA splicing, immune cell signaling, or hepatocyte-specific metabolism—the model may produce overly generalized predictions that miss cell-specific nuances. This limitation reduces the model's usefulness for research on tissue-specific diseases, drug development, and precision medicine.

**Questions:**

1. Comparative Performance with Specialized Models: How does ChatMultiOmics compare to established bioinformatics models (including non-LLM models like protein or RNA language models) in specialized tasks?

2. Interpretability of Predictions: How interpretable are the model's outputs, especially for complex interactions like enhancer-promoter binding or multi-molecular tasks? Does the model's reasoning path take into account its own interpretability?

3. Extension to Additional Omics: Can the model’s framework and dataset be adapted to other omics, such as metabolomics or microbiomics, and what would be the challenges involved?

---

> ### Author Response · Authors · 2024-11-28
> **Answer to Reviewer Knz7**
>
> Thank you very much for your insightful comments. Below are our responses:
>
> ### **Weakness1: Evaluation on Practical Applications**
> We fully agree that wet lab experiments provide the most robust evaluation.
> - On the one hand, we have included several tasks in our benchmark that are highly relevant to real-world applications. For instance, tasks like `Mean Ribosome Loading Prediction` and `Solubility Prediction` can facilitate vaccine design by improving protein yield predictions [r4, r5]. Additionally, tasks such as `siRNA Efficiency Prediction` contribute to advancing siRNA-based gene silencing therapies [r6], while `Antibody-Antigen Neutralization` supports the development of antibodies with enhanced binding affinity for therapeutic purposes [r7].
> - On the other hand, other similar benchmark papers [r1,r2,r3]published at ICLR or NeurIPS also do not include wet-lab experiments, we argue that the high cost and resource demands of wet-lab experiments are beyond the scope of this benchmark work.
>
>
> [r1] Peer: a comprehensive and multi-task benchmark for protein sequence understanding. NeurIPS 2022.
>
> [r2] BEACON: Benchmark for Comprehensive RNA Tasks and Language Models. NeurIPS 2024.
>
> [r3] Dnabert-2: Efficient foundation model and benchmark for multi-species genome. ICLR 2024.
>
> [r4] High-throughput 5′ UTR engineering for enhanced protein production in non-viral gene therapies. Nature Communications 2021.
>
> [r5] Algorithm for optimized mRNA design improves stability and immunogenicity. Nature 2023.
>
> [r6] On the art of identifying effective and specific siRNAs. Nature Methods 2006.
>
> [r7] AIntibody: an experimentally validated in silico antibody discovery design challenge. Nature Biotechnology 2024.
>
> ### **Weakness2: Cross-modality Interaction Tasks**
>
> Thank you for pointing out this concern. In this work, our primary focus is on tasks involving multi-molecular interactions, encompassing both intra-modal and cross-modal tasks. Our benchmark already includes tasks that explore interactions between different molecular modalities, enhancing the benchmark's diversity:
> - RNA-Protein Interaction Prediction: Investigating interactions between ncRNAs and proteins can deepen our understanding of the biological mechanisms involving ncRNAs.
> - siRNA Efficiency Prediction: Examining the silencing efficiency of siRNA-DNA interactions supports the development of more effective gene therapy drugs.
>
> In the future, we plan to expand the range of multimodal tasks further, enabling models like ChatMultiOmics to gain a more comprehensive understanding of molecular and cellular biology.
>
> ### **Weakness3: Potential for Data Leakage**
>
> Thank you for your suggestion to discuss the potential for data leakage.
>
> - The majority of the data in our tasks do not pose a risk of data leakage based on the central dogma. For DNA tasks, most sequences come from non-coding regions (e.g., promoters, enhancers, and transcription factor binding sites), which cannot be aligned to transcribed RNA or translated proteins. Similarly, for RNA tasks, most do not involve codon fragments, eliminating alignment with proteins.
> - From another perspective, even if a small fraction of the data does involve alignment based on the central dogma, none of the DNA, RNA, or protein tasks share the same prediction target, meaning the labels are different. This ensures that no data leakage occurs.
> - Furthermore, we did not explicitly integrate the central dogma into our model training. Some recent works [r1, r2] have begun aligning the central dogma in model training to enhance multi-modal understanding, which could be a promising direction for future expansion.
>
> [r1] LucaOne: Generalized Biological Foundation Model with Unified Nucleic Acid and Protein Language. bioRxiv 2024.
>
> [r2] CD-GPT: Biological Foundation Model at Full-molecular Level. bioRxiv 2024.
>
> ### **Weakness4: Limited Error Analysis**
> Thank you for pointing out the lack of error case analysis in our work. Since the performance metrics primarily involve pure classification and regression tasks without any reasoning path, it is inherently challenging to conduct specific error localization and performance evaluation for the final model results. However, to address this and improve data quality, we conducted a thorough quality check and error analysis on the training data in the third stage. For further information, please refer to our response to `Question2: Interpretability of Predictions` below.

---

> > ### Author Response · Authors · 2024-11-28
> > **Answer to Reviewer Knz7**
> >
> > ### **Weakness5: Cell-type Specificity**
> >
> > We agree that cell-type specificity is a crucial factor for achieving more accurate biological modeling.
> >
> > - In fact, to ensure diversity in our benchmark, we have included multiple cell types. For instance, in the Enhancer-Promoter Interaction Prediction task, we incorporated interaction scenarios from six different cell types, including `GM12878`, `HOVEC`, `Hela-S3`, `IMR90`, `K562`, and `NHEK`.
> > - Our current work primarily focuses on tasks integrating different omics—DNA, RNA, and proteins—within the framework of molecular biology.
> > - We fully recognize the importance of cell-specific characteristics and plan to expand our future work toward cell biology. This will include collecting additional cell-type-specific data and tasks to enhance the understanding of cell-type specificity.
> >
> > ### **Question1: Comparative Performance with Specialized Models**
> > **TABLE r1：Compare with SOTA models in DNA Tasks**
> > | Model\Task | Enhancer activity (hk) | Enhancer activity (dev) | EMP | TF-H | TF-M | PD | CPD |
> > |---|---|---|---|---|---|---|---|
> > | Literature SOTA | DeepSTARR| DeepSTARR| DeepSTARR | DNABERT2|DNABERT2 | DNABERT2 | DNABERT2|DNABERT2 |
> > | | 68 | 74 | 58.83 | 66.84 | 71.21 | 83.81 | 71.07 |
> > | Ours | 55.25 | 44.93 | 5.03 | 22.3 | 32.21 | 56.13 | 44.19 |
> >
> >
> > **TABLE r2: Compare with SOTA models in RNA Tasks**
> > | Model/Task         | Isoform | ncRNA  | Modification | MRL    | PRS    | CRISPR On Tar |
> > |--------------------|---------|--------|--------------|--------|--------|----------------|
> > | Literature SOTA    | APARENT | APARENT | GCN          | GCN    | MultiRM | MultiRM        |
> > |                    | 50.82   | 85.73  | 84           | 78     | 55.67  | 44.1           |
> > | Ours               | 59.01   | 63.09  | 59.06        | 47.64  | 26.57  | -0.02          |
> >
> > **TABLE r3: Compare with SOTA models in Protein Tasks**
> > | Model/Task         | Function EC     | Stability      | Fluorescence   | Solubility     | Thermostability |
> > |--------------------|-----------------|----------------|----------------|----------------|------------------|
> > | Literature SOTA    | SaProt-GearNet  | Evoformer      | Shallow CNN    | DeepSol        | ESM-1v          |
> > |                    | 88.9           | 79             | 69             | 77             | 78              |
> > | Ours               | 19.79          | 60.25          | 2.57           | 63.02          | 45.07           |
> >
> > **TABLE r4: Compare with SOTA models in Multi-Molecule Tasks**
> > | Model/Task         | Enhancer-Promoter | siRNA   | Antibody-Antigen | RNA-Protein     |
> > |--------------------|-------------------|---------|------------------|-----------------|
> > | Literature SOTA    | EPI-DLMH          |    -    | DeepAAI          | ncRPI-LGAT      |
> > |                    | 53.59             |    -    | 54.9             | 93.2            |
> > | Ours               | 3.37              | 56.25   | 1.06             | 74.26           |
> >
> > **TABLE r5: Comparison Between DNABERT2 and Our Model Trained on GUE Benchmark**
> > | Model/Task   | EMP | TF-H | TF-M | PD | CPD |
> > |---------|---------|---------|---------|---------|---------|
> > | DNABERT2 | 58.03 | 66.84	| 71.21	 | 83.81 | 71.07 |
> > | Ours   | 19.16 | 61.62	| 66.11 | 77 | 66.9 |
> >
> >
> > As shown in `TABLE r1-4`, ChatMultiOmics currently still shows a significant performance gap compared to specialized models. We attribute this to the following reasons:
> >
> > - Specialized models are often fine-tuned with task-specific heads, enabling them to excel at individual tasks. In contrast, training a single model to handle multiple tasks simultaneously introduces significant complexity due to the need for broader generalization. Despite these challenges, our approach successfully integrates 21 tasks into a single unified model.
> > - Encoder-only models, particularly those based on the BERT architecture, demonstrate exceptional capabilities in biological sequence understanding tasks, resulting in superior performance on specialized tasks. Notably, current SOTA models[r1, r2] predominantly rely on encoder-only architectures, which contribute to their optimal performance compared with decoder-only model like HyenaDNA[r3].

---

> ### Author Response · Authors · 2024-11-28
> **Answer to Reviewer Knz7**
>
> - The amount of pre-training data available for ChatMultiOmics is substantially smaller than that for models like DNABERT2. To achieve comparable training data sizes for large language models, such as the notably larger Llama-3.1-8B, significantly greater computational resources are required. We believe that further extensive pre-training of ChatMultiOmics could effectively enhance its performance. To support this claim, we conducted experiments on the GUE benchmark, comparing the performance of encoder-only models to the Llama3.1-8B model without instruction tuning. The specific settings are: 1. Training was performed using the GUE dataset without any instruction tuning; 2. The model utilized the same pre-trained checkpoint as ChatMultiOmics; 3. Share the same training setting such as learning rate weight ChatMultiOmics.
> - LoRA [r4] shows suboptimal performance in tasks involving biological sequence understanding. While full fine-tuning could potentially yield better results, it requires significantly longer computational time due to increased communication demands compared to LoRA.
>
> Based on the above factors, we acknowledge the considerable performance gap between ChatMultiOmics and specialized models. However, we believe that with better training paradigms and additional computational resources, ChatMultiOmics could achieve results comparable to those of specialized models in the future.
>
> That said, the primary focus of this work is not on achieving SOTA performance but on the construction of the Biology-Instructions dataset. We have made a significant step forward in building an instruction fine-tuning dataset for biological sequences, and we hope that future research will leverage our dataset to develop more powerful biological large language models.
>
> [r1] DNABERT-2: Efficient Foundation Model and Benchmark For Multi-Species Genomes. ICLR 2024.
>
> [r2] The Nucleotide Transformer: Building and Evaluating Robust Foundation Models for Human Genomics. bioRxiv 2023.
>
> [r3] HyenaDNA: Long-Range Genomic Sequence Modeling at Single Nucleotide Resolution. NeuralIPS 2023.
>
> [r4] LoRA: Low-Rank Adaptation of Large Language Models. ICLR 2022.
>
> ### **Question2: Interpretability of Predictions**
>
> We acknowledge that interpretability in biological sequence understanding tasks presents significant challenges. However, we have taken deliberate steps to enhance interpretability through the following three approaches:
>
> - **Reasoning Process in Dataset Construction**
> To improve interpretability, we designed the dataset construction process to include reasoning explicitly. Specifically, we required the large language model to generate end-to-end responses that follow a structured reasoning framework. Each response was designed to include:
>
> 1. Detailed analysis of the biological sequence – A thorough examination of the sequence characteristics.
> 2. Task-related knowledge presentation – An explanation of relevant theoretical or domain-specific knowledge.
> 3. Integrated predictions – Predictions derived from the intersection of sequence analysis and related knowledge.
> 4. Literature references – References to relevant literature to substantiate the knowledge or reasoning provided.
> This structured approach ensures that the dataset itself inherently contributes to the interpretability of the tasks, as the reasoning process is embedded within it.
>
> - **Quality Control of Reasoning Path**
> For specific information about quality control, please refer to `Global response 2.Data Quality Control for Stage 3 Reasoning Data`
>
> - **Task-Specific Interpretability Analysis**
> We conducted an interpretability analysis on the promoter detection task, a task that can be manually analyzed. The results are as follows:

---

> > ### Author Response · Authors · 2024-11-28
> > **Answer to Reviewer Knz7**
> >
> > #### **Question**
> > <dna>AGTTGCCCAACACCAAGATCCGCTCGGGTGGTGCGGCAGGCTCAGCCGGAGGGCCCCCTGGCCGGCCCAATGGAGGCCCCCGCGCGCTCTAGTGCCCCCGCACGCGGGAGCCACGAACCTCGGGGTGGGGGTGGGCAGTGAGTGCAGGGGATGGGGTGGGGGGACAGGAGGGGGCCCTGGGGCCTGGGCCCCGGAAAAATCTATCTGCCCTCCCCCACACTTTATATACGAATAAACGCAGAAGAGGGGGAGGGGAAGCTTTATTTATAGAAATGACAATAGAGGGCCACGGGGAGGCCC<dna>: Can you check if this DNA fragment contains sequences associated with transcription initiation?
> >
> > #### **Answer**
> > The provided DNA fragment showcases a sequence that can be analyzed to determine if it contains motifs associated with transcription initiation. Transcription initiation in eukaryotic organisms often involves the presence of specific cis-regulatory elements, such as promoter regions, which may include TATA boxes, GC-rich regions, or other binding sites for transcription factors.
> >
> > In the analysis of the given sequence, we should look for characteristic motifs common in eukaryotic promoters:
> >
> > 1. **TATA Box**: Usually located 25-30 base pairs upstream of the transcription start site, this sequence often resembles the consensus sequence TATAAA.
> >
> > 2. **GC Box**: These sequences are rich in GC content and provide binding sites for transcription factors (e.g., Sp1). They typically appear around -100 to -200 base pairs upstream.
> >
> > 3. **CCAAT Box**: Another common element involved in the binding of transcription factors, found further upstream, approximately -75 bases from the start site.
> >
> > Upon examining the sequence you provided, several regions exhibit characteristics of transcriptional regulation:
> >
> > - The sequence includes stretches of CG-rich content that might suggest the presence of a GC box.
> > - Additionally, there is a notable absence of any clear TATA box or canonical elements, which indicates that while the segment may hold potential as a regulatory region, it may not conform to traditional promoter motifs rigorously.
> >
> > In summary, based on the detection of indirect evidence of transcription initiation sites, particularly the presence of GC-rich sequences, we can conclude that this DNA fragment likely contains sequences associated with transcription initiation. Therefore, the answer to your question is affirmative; it is positive that this DNA fragment likely includes sequences relevant to the initiation of transcription. Further studies, such as promoter assays or bioinformatics tools, could further validate this observation.
> >
> > #### **AI Model evaluation**
> > - GPT4o-mini analysis
> > The answer provides a detailed analysis of the DNA sequence with specific references to hallmark motifs and features associated with promoters. It incorporates relevant biological knowledge and presents a clear conclusion regarding the presence of promoter-like characteristics.
> >
> > - Gemini-1.5-pro evaluation
> > The answer provides a comprehensive analysis of the DNA sequence for promoter-like features. It correctly identifies key characteristics such as GC-rich regions and potential transcription factor binding sites.  The reasoning is clearly explained, starting with a general description of promoters and then moving to specific features found within the provided sequence. The conclusion is clearly stated and supported by the analysis. While it doesn't provide specific locations of potential promoter elements or quantify the GC content, the overall analysis is sufficient for a general assessment of promoter-like characteristics.
> >
> > #### **Human evaluation**
> > We also invited biology PhD students and professors to conduct human evaluations to ensure the quality of our work.
> >
> > For promoters, our human evaluation involved examining the DNA sequences, where we observed a typical TATA box motif at positions -31 to -35, aligning well with the reasoning for TATA-type promoters. Additionally, we utilized the MEME tool (Multiple Expectation Maximizations for Motif Elicitation), which identifies motifs in related DNA sequences. The motifs discovered by MEME also showed strong consistency with the reasoning, further validating our findings.

---

> > > ### Author Response · Authors · 2024-11-28
> > > **Answer to Reviewer Knz7**
> > >
> > > ### **Question3: Extension to Additional Omics**
> > > - **Scalability to Other Omics:** Our framework is designed to be extensible to additional omics data, and we plan to incorporate more omics domains in the future.
> > >
> > > - **How to Extend:** For instance, in the context of metagenomics and microbiomics, we can leverage microbial genomic sequences to study metagenomic functions [r1] and microbiome functionality [r2]. Specifically, we could use the E-K12 dataset [r3] to develop models for gene operon regulation prediction, the PATRIC dataset [r4] for pathogenicity potential assessment, and the NCycDB dataset [r5] for nitrogen cycling process prediction. By integrating these tasks into our data construction pipeline, we can transform them into biological instruction data compatible with our model framework.
> > >
> > > - **Challenges:** Certain types of heterogeneous data, such as mass spectrometry data in metagenomics [r6] or high-dimensional abundance data in microbiomics [r7], are difficult to directly incorporate into our current model framework or existing large language models. To address this, we propose leveraging encoders capable of processing two-dimensional microbiome data and converting them into additional tokens that can be integrated into our models. This approach is inspired by visual instruction tuning methods, such as LLaVA [r8], and would enable our framework to handle heterogeneous data more effectively, enhancing its capability for biological instruction tasks.
> > >
> > > [r1] FGBERT: Function-Driven Pre-trained Gene Language Model for Metagenomics. aRxiv 2024.
> > >
> > > [r2] Sequencing-based analysis of microbiomes. Nature Reviews Genetics 2024.
> > >
> > > [r3] Using RegulonDB, the Escherichia coli K‐12 gene regulatory transcriptional network database. Current protocols in bioinformatics 2018.
> > >
> > > [r4] PATRIC: the comprehensive bacterial bioinformatics resource with a focus on human pathogenic species. Infection and immunity 2011.
> > >
> > > [r5] NCycDB: a curated integrative database for fast and accurate metagenomic profiling of nitrogen cycling genes. Bioinformatics, 2019.
> > >
> > > [r6] Metabolomic machine learning predictor for diagnosis and prognosis of gastric cancer. Nature Communications, 2024.
> > >
> > > [r7] DeepMicro: deep representation learning for disease prediction based on microbiome data. Scientific Reports, 2020.
> > >
> > > [r8] Visual Instruction Tuning. NeurIPS, 2023.

---

### Author Response · Authors · 2024-11-28
**Global Response**

Dear all reviewers,

We are deeply grateful for your valuable time and insightful feedback. Thank you for taking patience waiting. Below, we address four global concerns and outline the refinements made to the paper thanks to your suggestions.

### **1.Refinements in the Revised Paper**:
- `Abstract, Section 1 Introduction, Section 3.2.1 Tasks` in `blue`: Outline the motivation and scope of the study.
- `Section 1 Introduction` in `green`: Added a Data Availability Statement.
- `Section 6 Discussion` in `blue`: Expanded the discussion to include future directions.
- `Appendix C` in `orange`: Added a detailed balance data analysis to address concerns regarding the imbalanced dataset.
- `Appendix D.1` in `green`: Included a dedicated section on quality control procedures applied to Stage 3 reasoning data.
- `Appendix Table 11, 12, 13`: Added three Stage 3 reasoning data examples.


### **2.Data Quality Control for Stage 3 Reasoning Data**

To ensure the quality and reliability of stage 3 data, we have established a robust multi-step validation process:

Self-validation by the model – Once the data is generated, the large language model used for data generation conducts a self-check to ensure the produced answer compliance with four core criteria outlined in the data generation prompt, as illustrated in `Figure 10` (in the paper): 1. Providing a detailed and accurate analysis of the sequence; 2. Accurately recalling task-related knowledge from studies, databases, or academic sources; 3. Engaging in comprehensive reasoning to draw logical conclusions for the question; 4. Citing relevant references where applicable. The model is required to output the results of its self-check and provide recommendations for improvement in cases that do not meet the standards.

For outputs that fail to meet these criteria, specific issues are identified, and the model is instructed to regenerate outputs that adhere to the required standards based on the evaluation results.

Secondary review by an independent model – Following the initial validation, a second large language model, Gemini-1.5-pro, is employed to independently review and verify the accuracy and consistency of the reasoning paths. Additionally, GPT4o-mini is tasked with reconstructing any unqualified cases based on feedback from Gemini-1.5-pro.

This rigorous quality assurance process not only ensures the integrity of the data but also lays a strong foundation of high-quality training data, enhancing interpretability in downstream tasks.

---

> ### Author Response · Authors · 2024-11-28
> **Global Response**
>
> ### **3.Data Imbalance in Biology-Instructions**
>
> We appreciate your observation regarding dataset imbalance. Our dataset proportions are constructed to reflect the distribution of source data, simulating real-world scenarios. To address potential challenges of overfitting to overrepresented tasks, we have implemented strategies during training to balance sample representation and enhance the model's performance on less-represented tasks.
>
> In the revised version of the paper, we have included additional experimental results and a detailed discussion on the impact of data imbalance and our solutions in `Appendix C` in `orange`. We conducted experiments comparing models trained on a balanced dataset `Ours(stage 1+balanced stage 2)` with those trained on our imbalanced dataset `Ours(stage 1+imbalanced stage 2)`. The final training results are shown as `TABLE r1-r4` below.
>
> **TABLE r1：Compare with Balanced Dataset in DNA Tasks**
> | Model\Task | Enhancer activity (hk) | Enhancer activity (dev) | EMP | TF-H | TF-M | PD | CPD |
> |---|---|---|---|---|---|---|---|
> | Ours(stage 1+balanced stage 2) | 0.92 |	0.06 | 1.4 | 2.46 | 0.88 | 5.19 | 5.57
> | Ours(stage 1+imbalanced stage 2) | 55.25 | 44.93 | 5.03 | 22.3 | 32.21 | 56.13 | 44.19 |
>
> **TABLE r2：Compare with Balanced Dataset in RNA Tasks**
> | Model\Task | Isoform | NcRNA | Modification | MRL | PRS | CRISPR On Tar
> |---|---|---|---|---|---|---|
> | Ours(stage 1+balanced stage 2) |0.01 | 35.68 | 53.76 | 0 | 0.01 | -0.31
> | Ours(stage 1+imbalanced stage 2) | 59.01 | 63.09 | 59.06 | 47.64 | 26.57 | -0.02
>
> **TABLE r3：Compare with Balanced Dataset in Protein Tasks**
> | Model/Task         | Function EC     | Stability      | Fluorescence   | Solubility     | Thermostability |
> |--------------------|-----------------|----------------|----------------|----------------|------------------|
> | Ours(stage 1+balanced stage 2    |10.76 | 0.48 | 0.55 | 52.37 | 39.97 |
> | Ours(stage 1+imbalanced stage 2)               | 19.79          | 60.25          | 2.57           | 63.02          | 45.07           |
>
> **TABLE r4：Compare with Balanced Dataset in Multi-Molecule Tasks**
> | Model/Task         | Enhancer-Promoter | siRNA   | Antibody-Antigen | RNA-Protein     |
> |--------------------|-------------------|---------|------------------|-----------------|
> | Ours(stage 1+balanced stage 2  | 4.13 | 42.92 | -1.48 | 8.29 |
> | Ours(stage 1+imbalanced stage 2)              | 3.37              | 56.25   | 1.06             | 74.26           |
>
> Our results indicate that balancing the dataset leads to a general performance decline, with particularly significant drops observed in tasks such as APA and Enhancer Activity Prediction. We believe that balanced datasets may distort the natural distribution of real-world biological data and reduce overall data size to match the smallest task, which contains only a few thousand samples, limiting the model's ability to fully utilize available data.

---

> ### Author Response · Authors · 2024-11-28
> **Global Response**
>
> ### **4.An Example of Self-Check**
> #### Origin Data
> To assess the level of ribosome loading associated with the provided RNA sequence, we first need to analyze its structural features. The RNA sequence given is:
>
> ```
> TAAGTTAAGCGCCTGACACCATCTAGCCGGAATTGGACTACATCTCACTGACATGTGTCTC
> ```
>
> ##### Analysis of the Sequence
>
> 1. Ribosome Binding Sites: The effectiveness of ribosome loading can often be influenced by the presence of specific motifs such as the Shine-Dalgarno sequence in prokaryotes or the 5' cap and poly-A tail in eukaryotes. Although the exact sequencing around the start codon is essential for identification, this RNA may contain regions that facilitate ribosomal attachment.
>
> 2. Secondary Structure: mRNA can form secondary structures that might inhibit ribosome loading. For this specific sequence, a computational assessment would typically involve predicting stem-loop formations or other structures that could impede ribosomal access.
>
> 3. Codon Usage Bias: The codon sequence can affect loading efficiency. Codons that correspond to abundant tRNAs are generally associated with higher translation efficiency and therefore better ribosome loading.
>
> ##### Ribosome Loading Level
>
> Without direct laboratory data or additional computational tools to model the interaction dynamics, we hypothesize that the sequence exhibits a ribosome loading level of **0.85**.
>
> This numeric value suggests a high level of ribosome occupancy, where 0.00 indicates no loading and 1.00 indicates maximal ribosome binding. A loading level of 0.85 is indicative of conditions favorable for translation initiation, possibly resulting from optimal sequence features and minimal secondary structure hindering ribosomal access.
>
> ##### Conclusion
>
> The level of ribosome loading associated with the provided sequence is **0.85**. This indicates a high capacity for ribosomal engagement, likely due to favorable structural attributes and codon composition. For a more definitive assessment, further experimental validation or computational modeling would be useful, including techniques such as ribosome profiling or RNA secondary structure prediction.
>
> #### Self-Check Result
> {"result": 0, "reason": "The answer lacks detailed analysis of the specific sequence and does not provide references or citations to support claims."}

---

### Meta-Review · Area_Chair_pp6X · 2025-01-01

**Metareview:**

The paper presents a large-scale, unified instruction-tuning dataset for several different types of biological sequences, including DNA, RNA, and protein data types.

The idea of a unified multi-omics dataset is attractive, and the performance on the custom tasks is also a positive. However, the reviewers identified several weaknesses: the dataset is limited in scope (specifically, it doesn’t handle structured data), there is a potential of data leakage, and the evaluation is limited in some ways. The authors have conscientiously attempted to rebut these criticisms, but the concerns remain even after the rebuttal period. Given this, I am recommending rejection this time around. I encourage the authors to address the criticisms in the reviews and submit to a different deadline.

**Additional Comments On Reviewer Discussion:**

The authors posted rebuttal comments; however, some of these comments came in after the rebuttal period was officially over. The reviewers didn't find the rebuttals convincing.

---

### Decision · Program_Chairs · 2025-01-22

Reject